# Demystifying Scientific Problem-Solving in LLMs by Probing Knowledge and Reasoning

Alan Li [* 1]   Yixin Liu [* 1]   Arpan Sarkar [2]   Doug Downey [3 4]   Arman Cohan [1 3]

## Abstract

Scientific problem solving poses unique challenges for LLMs, requiring both deep domain knowledge and the ability to apply it through complex reasoning. While automated scientific reasoners hold great promise for assisting human scientists, there is currently no widely adopted holistic benchmark for evaluating scientific reasoning, and few approaches systematically disentangle the distinct roles of knowledge and reasoning in these tasks. To address these gaps, we introduce SCIREAS, a diverse suite of existing benchmarks for scientific reasoning tasks, and SCIREAS-PRO, a selective subset that requires more complex reasoning. Our holistic evaluation surfaces insights about scientific reasoning performance that remain hidden when relying on individual benchmarks alone. We then propose KRUX, a probing framework for studying the distinct roles of reasoning and knowledge in scientific tasks. Combining the two, we conduct an in-depth analysis that yields several key findings: (1) Retrieving task-relevant knowledge from model parameters is a critical bottleneck for LLMs in scientific reasoning; (2) Reasoning models consistently benefit from external knowledge added in-context on top of the reasoning enhancement; (3) Enhancing verbalized reasoning improves LLMs' ability to surface task-relevant knowledge.[1]

## 1. Introduction

Recent frontier reasoning models, such as OpenAI's o-series (OpenAI et al., 2024) and DeepSeek-R1 (Guo et al., 2025), demonstrate significant advances by leveraging increased test-time compute to enable intermediate reasoning steps (Wei et al., 2023; Kojima et al., 2023). These approaches facilitate advanced mechanisms, including methodology exploration (Yao et al., 2023), self-verification (Ma et al., 2025a), and backtracking (Yang et al., 2025), resulting in improvements on tasks such as mathematics and coding (Muennighoff et al., 2025).

These advances in reasoning capabilities create opportunities for applying LLMs to complex scientific tasks (Lu et al., 2024; Gottweis et al., 2025; Schmidgall et al., 2025). However, scientific work demands rigorous reasoning and deep domain knowledge, from specialized concepts and foundational theories to hands-on methodological expertise and familiarity with obscure yet pivotal findings. Successful scientific reasoning systems must apply such knowledge in complex reasoning processes (Lu et al., 2022; Luo et al., 2025; Wadden et al., 2025; Li et al., 2025).

While a variety of scientific benchmarks exist (e.g., GPQA (Rein et al., 2024) and MMLU-Pro (Wang et al., 2024b)), there is no holistic and unified benchmark that comprehensively targets scientific reasoning. Existing individual benchmarks typically focus narrowly on specific domains, task formats, or skill types. For example, although GPQA is challenging, it focuses exclusively on multiple-choice questions within a limited range of domains. Furthermore, there is a lack of analytical tools that can isolate the distinct roles that reasoning and scientific knowledge play when performing sophisticated scientific tasks.

We introduce datasets and methods to facilitate the study of scientific problem solving. First, we present SCIREAS, a unified suite of 10 public benchmarks that span physics, chemistry, biology, medicine, materials, mathematics, computer science, and engineering, with multiple-choice, fill-in-the-blank, structured, and protocol/procedural questions. To improve evaluation efficiency and sharpen the focus on reasoning difficulty, we manually inspect each subtask and retain only those that are subject-relevant and

---
[*]Equal contribution [1]Department of Computer Science, Yale University [2]Department of Molecular & Cellular Biology, Harvard University [3]Allen Institute for AI [4]Department of Computer Science, Northwestern University. Correspondence to: Alan Li <haoxin.li@yale.edu>, Yixin Liu <yixin.liu@yale.edu>.

*Proceedings of the 43rd International Conference on Machine Learning*, Seoul, South Korea. PMLR 306, 2026. Copyright 2026 by the author(s).

[1]The codebase and artifacts are released at https://github.com/yale-nlp/SciReas-Eval.

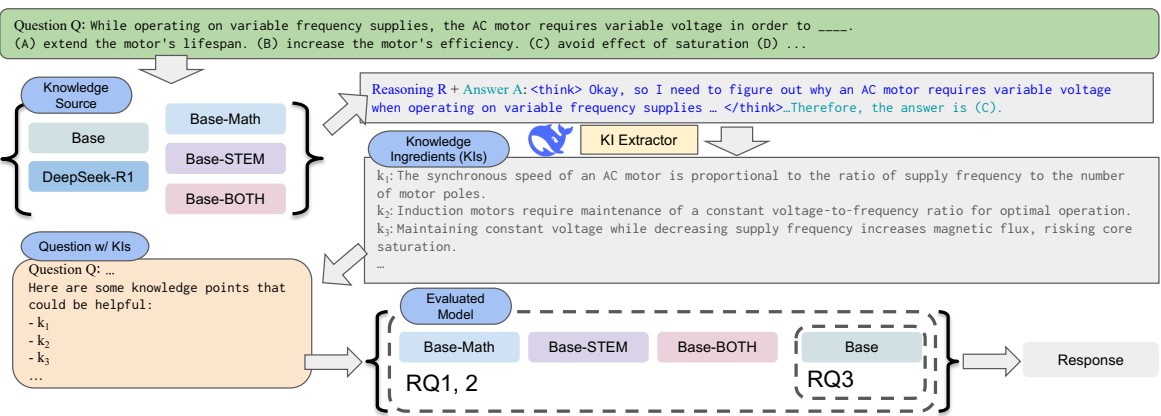

*Figure 1.* KRUX pipeline. Starting upper left, we prompt an LLM (one of base, DeepSeek-R1, Base-Math, Base-STEM, and Base-BOTH) with a question from SCIREAS as the knowledge source, collect the output and reasoning traces, and feed the reasoning traces to DeepSeek-R1 as the extractor to generate knowledge ingredients (KIs). We then evaluate the tested model with KI-augmented questions, which allows us to study three key research questions (RQ1 §4.3, RQ2 §4.4, RQ3 §4.5) regarding LLMs' knowledge and reasoning capabilities in scientific problem-solving.

reasoning-intensive, while preserving broad domain coverage. Furthermore, we provide an efficient, unified implementation that standardizes evaluation across benchmarks, eliminating the need for separate environments or dataset-specific boilerplate (§3.1).

Next, we introduce **SCIREAS-PRO**, a compact subset of SCIREAS tailored for evaluating more challenging reasoning. Specifically, SCIREAS-PRO is constructed by selecting examples from SCIREAS where only reasoning models with high inference compute budget (or the highest allowed number of thinking tokens) succeed. We find that despite containing only 8% as many examples as SCIREAS, SCIREAS-PRO better differentiates weak and strong reasoners (§3.1).

Having constructed the reasoning-intensive scientific benchmarks, our next goal is to leverage them to study how verbalized chain-of-thought (CoT) reasoning affects knowledge recall and usage (§4). To study this, we design **KRUX** (Knowledge & Reasoning Utilization eXams), a probing framework which supplies models with atomic "knowledge ingredients" (KIs) extracted from other models' reasoning traces (Figure 1). This technique allows for more controlled analyses of reasoning and knowledge, which we use to perform investigations that lead to the following findings:

(1) Vanilla instruct models can *outperform* their reasoning-fine-tuned counterparts by $\geq$ 10% once KIs are provided in-context, **suggesting that internalizing and retrieving the right knowledge is a key bottleneck for scientific reasoning tasks.**

(2) When both model families receive the same KIs from a strong reasoner (e.g., DeepSeek-R1), the reasoning-fine-tuned models consistently outperform their base models, showing that **reasoning models are capable of utilizing**

**external knowledge for additional improvements.**

(3) Feeding KIs from a reasoning-fine-tuned model to its base model can boost performance even when the KIs are already known by the base model, indicating that **reasoning-fine-tuning aids knowledge recall by surfacing more relevant knowledge**.

Our contributions can be summarized as:

• We introduce SCIREAS, a unified and holistic benchmark suite spanning a broad range of scientific domains and problem types, allowing us to surface insights that otherwise remain hidden if relying on individual datasets only. The reasoning-focused subset SCIREAS-PRO allows efficient benchmarking of sophisticated reasoning with more room for improvement.
• We present KRUX, a novel analytic framework which we use to conduct a comprehensive empirical study that disentangles the impacts of knowledge and reasoning.
• We provide an in-depth analysis with three key findings: (i) knowledge retrieval is a bottleneck; (ii) in-context knowledge consistently benefits reasoning models; and (iii) long CoT improves knowledge surfacing. We support these findings with controlled post-training experiments and show our training recipe is competitive compared with concurrent SFT post-training efforts.

## 2. Related Work

**Scientific Benchmarks** Existing scientific benchmarks span a wide array of domains and tasks, but each tends to focus on specific disciplines or subskills, often lacking explicit emphasis on multi-step reasoning or standardized implementation. For example, most tasks in SciRIFF (Wadden et al., 2025) focus on context-grounded information QA, rather

than demanding reasoning. Benchmarks like GPQA (Rein et al., 2024) and LabBench (Laurent et al., 2024) pose reasoning challenges, yet they cover only a limited range of scientific domains and rely on multiple-choice QA formats. Benchmarks like CURIE (Kon et al., 2025) and QASA (Lee et al., 2023) offer exposure to scientific subjects, but their long-context feature hinders our objective to disentangle the effect of knowledge and reasoning. Implementation-wise, benchmarks lack standardized prompts, up-to-date evaluation metrics, or consistent scoring and reporting, making reproducibility and fair comparison difficult (Gu et al., 2025; Sutawika et al., 2024). To address this fragmentation, our study systematically incorporates 10 prominent scientific benchmarks, GPQA, MMLU-Pro (Wang et al., 2024b), SuperGPQA (Team et al., 2025b), LabBench, Olympiad-Bench (He et al., 2024), SciBench (Wang et al., 2024a), SciRIFF, UGPhysics (Xu et al., 2025), SciEval (Sun et al., 2024), and SciKnowEval (Feng et al., 2025), enabling a unified, comprehensive, and reproducible evaluation suite of scientific reasoning capabilities.

**Knowledge & Reasoning** An important line of work on disentangling reasoning and knowledge designs specialized tasks (e.g., linguistically questions (Bean et al., 2024; Khouja et al., 2025) or synthetic multi-hop questions (Li & Goyal, 2025)) to isolate reasoning from knowledge, but such benchmarks are often artificial and domain-constrained. Notably, Li & Goyal (2025) analyzes the synergy between knowledge and reasoning as knowledge evolves, offering a perspective complementary to our controlled CoT SFT experiments. Another line of work trains external classifiers to label questions as reasoning- or knowledge-intensive based on parametric models (Thapa et al., 2025). However, this approach requires well-calibrated training data and does not distinguish the tested model's internal knowledge from reasoning. Concurrent work leverages reasoning traces to evaluate factual correctness (Wu et al., 2025), but focuses on surface-level factuality rather than genuine knowledge recall. Unlike prior work that trains external classifiers to label question types or checks surface factuality in traces, KRUX holds knowledge constant and varies the target model, isolating knowledge recall from reasoning ability without relying on heuristic difficulty tags. Additional related work is provided in Appendix A.

# 3. Benchmarking Knowledge-Intensive Scientific Reasoning

Given limited coverage in terms of domain, formats, or accessibility for individual benchmarks, SCIREAS solves this by merging ten datasets under one standardized harness, offering broad domain coverage and consistent evaluation.

## 3.1. Evaluation Suite Construction

**SCIREAS** SCIREAS is an evaluation suite focused on reasoning-intensive scientific tasks curated from 10 representative existing benchmarks. Through task-level filtering, SCIREAS reduces instance count by nearly 50% while preserving coverage, and, inspired by OLMES (Gu et al., 2025), provides a unified implementation optimized with vLLM (Kwon et al., 2023) and batch job APIs[2] for scalable, easy-to-use, and efficient evaluation.

Our curation prioritizes subtasks from each benchmark that demand not only specific domain knowledge but also complex, multi-step reasoning processes for resolution. For each subtask from each benchmark, we (1) select candidate tasks based on documentation in their associated manuscripts that describe each subtask's characteristics, which indicate its difficulty and reasoning intensity. After that, we (2) use a subtask-level exclusion protocol to retain only those tasks that require both domain knowledge and multi-step reasoning, and (3) remove subjects outside mainstream STEM (e.g., weapon science and textile engineering).[3] Our procedure involves 3 authors as annotators, where two authors decide jointly in the first round, and another author validates by conducting the filtering process independently, following the same selection policy. This two-fold annotation validation reached an agreement accuracy of 90.1%. Uniform sampling is applied only after this manual screening and knowledge-scope annotation stage, so it does not affect which subtasks are retained. We provide more details and explanations for our protocol and the subtasks we selected in Appendix B.1.

To keep evaluation cost-efficient, we uniformly sample 200 instances from each subtask sourced from high-cost benchmarks — MMLU-Pro, SciKnowEval, SciEval, and UG-Physics, which maintains similar evaluation outcomes (more in Appendix B.2) while reducing the cost by nearly 50% (from 29,604 to 15,567 total instances). Benchmarks affected by our filtering are marked with an asterisk (*); their scores are not directly comparable to those from prior work.

**SCIREAS-PRO** Although SCIREAS provides a uniform measurement for model performance on scientific reasoning tasks that nominally require scientific reasoning, the difficulty of individual instances is uneven: some can be answered with little deductive effort once the pertinent fact is recalled, as shown in an example in Figure 2.

To isolate the reasoning skill, we therefore curate a "hard"

---

[2] We provide batch job inference options for popular LLM providers, e.g., OpenAI, Anthropic, TogetherAI, and Gemini. Using batch APIs allows for up to 50% cost reduction.

[3] While this manual inspection can be subjective, it is based on the authors' graduate-level expertise. We provide more details and examples in Appendix B.1.

**Reasoning Intensive**

To solve the problem, we start by analyzing the decomposition of KClO3. The balanced equation for its decomposition is: [2, KClO3 \\rightarrow 2, KCl + 3, O2]. The molar mass of KClO3 … For 49 g of KClO3, the moles of KClO3 are: … From the stoichiometry of the reaction, 2 moles of KClO3 produce 3 moles of O2. Thus, the moles of O\u2082 produced are: … The metal is amphoteric and abundant in the Earth's crust, strongly suggesting it is aluminum (Al). The molar mass of Al is 27 g/mol, so the moles of Al are: …

**Less Reasoning Intensive**

Here's the analysis for each option: Option A: The central boron atom is bonded to three oxygen atoms, each connected… Option B: This molecule consists of a benzene ring fused with three furan rings, each contributing ketone groups. … Option D is the only molecule with **C3h symmetry**, as it possesses the necessary **C3 axis** and **c3h mirror plane** while lacking vertical mirrors.

*Figure 2.* An example pair of solutions that require varying reasoning intensity, where the top example is sampled from SCIREAS-PRO and the bottom is a filtered out example (§3.1). At the top, the progressive reasoning chain is highlighted. Whereas the example at the bottom emphasizes knowledge recall on each option with a simple elimination strategy.

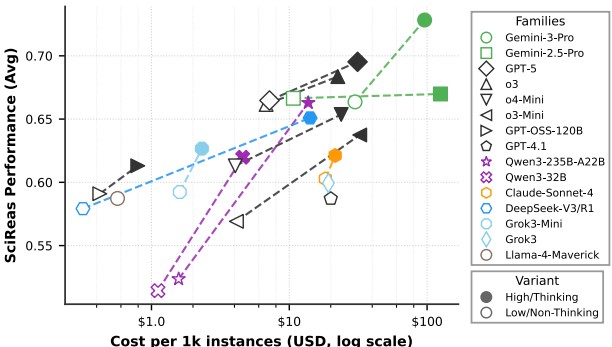

*Figure 3.* Frontier reasoning models' performance evaluated on SCIREAS. The X-axis shows the cost per 1k instances in USD. Different reasoning settings on the same model can result in distinct costs and performance, but the margins vary depending on the models.

subset — those questions whose solutions still demand multi-step inference even when all relevant knowledge is available — so that any performance gains cannot be explained by knowledge recall alone. Building on our observation in §3.2, we hypothesize that the performance difference under different test-time inference budgets can serve as an effective indicator of reasoning intensity. Specifically, instances where reasoning models *fail* with low reasoning budget but *succeed* with high budget likely require complex reasoning, even when the necessary domain knowledge is accessible to the model in both settings.

In practice, we evaluate o3-mini and o4-mini on SCIREAS with both *high* and *low* "reasoning-effort" settings, an OpenAI API flag that limits the number of thinking tokens before output. For o3-mini and o4-mini, the high-effort setting costs about 5.8× more per instance than the low-effort setting (Table 6, Appendix B.1).[4] For each model, we keep questions answered *incorrectly* under low effort but *correctly* under high effort and take the union of these sets to create SCIREAS-PRO, resulting in 1,260 unique instances. We further validate this approach by using LLM judge and human evaluation to check the reasoning-intensiveness of resulting examples from this filtering pipeline in Appendix B.3, and observe that incorrect answers are attributed to insufficient reasoning rather than lack of knowledge 90% of the time by humans on a sampled set and 91% of the time by LLM judge. We also construct an open-source-model variant, using Qwen3-32B thinking on/off as the selector; Appendix B.3.2 shows that it similarly amplifies low/high reasoning-effort gaps, suggesting the filtering is not specific

to proprietary selectors.

### 3.2. Benchmarking Frontier Models

Having constructed SCIREAS and SCIREAS-PRO with focus on scientific reasoning tasks, we now examine how frontier models perform under varying computational budgets. We evaluate frontier models using different "reasoning-effort" settings (see details in Appendix C). These settings typically correspond to significant differences in output length, with high-effort modes producing substantially more reasoning tokens as they work through complex problems.[5]

**Aggregated Results** Figure 3 highlights aggregated performance evaluated on SCIREAS, with score breakdowns on selected models shown in Table 6. Notably, the **aggregated ranking provides additional insights that differ from popular individual benchmarks**. Comparing o3-High and Gemini-2.5-Pro-Preview-High as an example, o3-High wins on GPQA and MMLU-Pro* while Gemini-2.5-Pro-Preview-High wins on SuperGPQA*, all with a thin margin (within 1 absolute point, even evaluated on MMLU-Pro before uniform sampling as shown in Figure 7). Similarly, GPT-5-High shows on-par performance with Gemini-2.5-Pro-Preview-High on problem-solving benchmarks like OlympiadBench and SciRIFF. Evaluated across SCIREAS, however, we notice that GPT-5-High outperforms its competitors on a broader range of benchmarks. Meanwhile, o3-High achieves higher overall performance over Gemini-2.5-Pro-Preview-High, with superior performance on LabBench* and weaker on OlympiadBench by a large margin (beyond 10 absolute points).

**Benchmark Correlations** In general, as the Pearson cor-

---

[4]Because these models are proprietary, factors beyond the flag may influence performance. We therefore treat the flag as a practical, not absolute, proxy and validate it with independent studies (Appendix B.3).

[5]In this work, we refer to DeepSeek-R1-0528 and DeepSeek-V3-0324 simply as DeepSeek-R1 and DeepSeek-V3, respectively, unless otherwise specified.

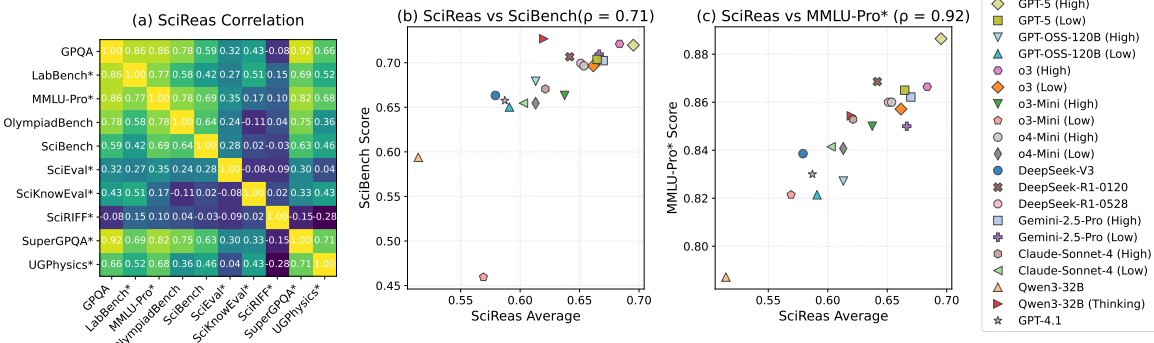

*Figure 4.* SCIREAS correlations breakdown. (a) Task-to-task Pearson correlations. SCIREAS incorporates tasks complementary to popular benchmarks. (b) and (c) show performance on SCIREAS vs. SciBench and MMLU-Pro*. Models may be tuned for certain tasks, outperforming higher-ranked models on individual benchmarks.

relations shown in Figure 4 (a), while some benchmarks are closely correlated (e.g., GPQA and SuperGPQA*), benchmarks containing free-form QA and fill-in-the-blank questions like SciRIFF* and SciEval* are not highly correlated with GPQA-like multiple-choice tasks, demonstrating the need for a holistic evaluation suite. Isolating specific benchmarks, we observe that **models from different providers may be tuned explicitly for specific tasks or skills**. As shown in Figure 4 (b) and (c), Qwen3-32B-Thinking strikes noticeably above the trend on SciBench, reaching comparable performance to commercial frontier models. Similarly, DeepSeek-V3 and DeepSeek-R1-0120 demonstrate stronger performance on MMLU-Pro*, indicating capabilities that surpass their overall rankings.

**Performance Gap by Reasoning Difference** Although the gap varies depending on different model families, **the same model can exhibit a significant performance gap under different reasoning settings**. For instance, in Figure 3, o3-mini-Low and -High show a performance gap of 6.8. Similar traits can be observed among o4-mini, Claude-Sonnet-4, and o3, while Gemini-2.5-Pro-Preview shows the least performance gain, even with significantly more ($>10\times$) thinking budget. This observation motivates the construction of SCIREAS-PRO, leveraging the performance gap between low and high reasoning efforts as an effective proxy for identifying instances that demand complex reasoning rather than mere knowledge recall. **For practitioners, task-specific evaluation is still recommended** for the optimal balance between inference cost and performance.

**Amplified Performance Gap** Figure 5 shows that SCIREAS-PRO **amplifies performance gaps between low- and high-reasoning settings**, where the gap between GPT-5-High and GPT-5-Low widens from 3.01 to 12.22, and the corresponding gap for Gemini-2.5-Pro-Preview widens from 0.35 to 2.30. Meanwhile, non-reasoning models, e.g., GPT-4.1 and DeepSeek-V3, exhibit larger gaps than concur-

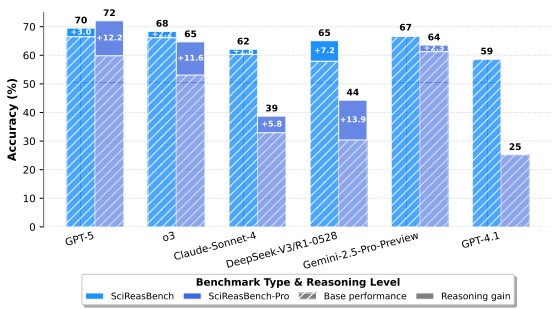

*Figure 5.* Model performance on SCIREAS and SCIREAS-PRO with varying reasoning capabilities. SCIREAS-PRO amplifies gaps between low-reasoning and high-reasoning settings.

rent reasoning models, o3 and DeepSeek-R1, respectively.

**Reasoning Efforts Improve Math Reasoning More** Is a higher inference budget more helpful to math or numeric reasoning than non-math reasoning? To answer this question, we categorize instances from SCIREAS into *Has-Math* and *No-Math* buckets (Appendix D.3.1) and report the gains in micro average accuracy. In Appendix D.3.2 Figure 10, the results show that higher reasoning budgets yield more improvements among Has-Math instances compared to No-Math instances. This finding echoes with concurrent work where Sprague et al. (2024) points out that CoT helps more with math and symbolic reasoning.

## 4. Disentangling Knowledge and Reasoning in Scientific Tasks

While SCIREAS and SCIREAS-PRO provide benchmarks to evaluate scientific reasoning capabilities, another fundamental question remains: how does CoT reasoning adaptation affect a model's ability to recall and utilize knowledge? To address this question, we first conduct a series of controlled SFT experiments on high-quality reasoning traces with and

Table 1. Performance of reasoning models trained from Qwen2.5-Instruct and Llama-3.1-Instruct on SYNTHETIC-1 and concurrent reasoning models.

| Model | Method | SCIREAS | -PRO |
|---|---|---|---|
| *Our Checkpoints* | | | |
| Qwen | – | 37.07 | 13.97 |
| Qwen-STEM | SFT | 40.47 | 16.11 |
| Qwen-Math | SFT | 41.99 | 18.17 |
| Qwen-BOTH | SFT | **42.84** | **21.11** |
| Llama | – | 31.25 | 11.67 |
| Llama-STEM | SFT | 35.28 | 14.29 |
| Llama-Math | SFT | 35.49 | 16.98 |
| Llama-BOTH | SFT | 38.55 | 16.51 |
| *Concurrent Reasoning Post-training* | | | |
| SYNTHETIC-1-SFT | SFT | 37.64 | 19.44 |
| OpenR1 | SFT | 43.08 | **26.43** |
| Llama-Nemotron | SFT&RL | **43.53** | 23.75 |
| General-Reasoner | RL | 34.99 | 13.73 |

without in-domain scientific knowledge, and then we propose KRUX, a novel investigative framework to study three key research questions regarding the role of knowledge in scientific reasoning using the fine-tuned checkpoints. In this setup, SCIREAS provides the unified scientific evaluation substrate, while KRUX is applied on top of these tasks as a diagnostic lens for knowledge recall and use.

## 4.1. Controlled CoT SFT

To control for data composition and isolate the impact of reasoning and knowledge injection during post-training, we fine-tune Qwen2.5-7B-Instruct (Qwen et al., 2025) and Llama-3.1-8B-Instruct (Grattafiori et al., 2024) on reasoning traces drawn from mathematics and STEM domains, as well as on their combination. This allows us to attribute behavior changes to the data mixture rather than confounding factors. Our goal is not to exhaustively compare post-training algorithms; rather, SFT provides a clean intervention for studying how reasoning traces and in-domain knowledge interact, while Table 1 contextualizes our checkpoints against concurrent SFT, RL, and SFT+RL models.

For training, we leverage the SYNTHETIC-1 (Mattern et al., 2025) dataset, an existing large-scale dataset released by Prime Intellect, which consists of outputs of DeepSeek-R1-0120, including the reasoning traces, on a diverse set of tasks. More specifically, we leverage the mathematics and STEM subsets from SYNTHETIC-1 (denoted as SYNTHETIC-1-Math/STEM, respectively). The former provides reasoning traces on abstract math questions, serving as a source for long CoT adaptation without introducing in-domain knowledge. In contrast, the latter is sourced from StackExchange (Lambert et al., 2023), providing a more in-domain data source for a broader range of scientific sub-

jects.[6] The math subset contains around 462K instances, while the STEM subset contains around 512K instances. Details of the training and evaluation setup are in Appendix D.

By training Qwen2.5-7B-Instruct on SYNTHETIC-1 (-Math, -STEM, and the combined subsets), we derived Qwen-Math, Qwen-STEM, and Qwen-BOTH along with their counterparts trained from Llama-3.1-8B-Instruct. In the following, we will refer to the Base models as Qwen or Llama for brevity. Compared with concurrent work on long CoT post-training (Bercovich et al., 2025; Hugging Face, 2025; Mattern et al., 2025; Ma et al., 2025b), our checkpoints deliver comparable performance under controlled settings (Table 1), serving as reliable investigating checkpoints. A lightweight analysis of domain-specific improvements and data composition is presented in Appendix D.3-D.4.

## 4.2. Knowledge & Reasoning Utilization Exam (KRUX)

We introduce KRUX (Figure 1), a novel investigative framework to study the role of knowledge and long CoT reasoning in scientific problem solving. To separate what a model *knows* from how it *reasons*, we hold knowledge availability fixed by injecting compact, answer-agnostic knowledge ingredients (KIs) in-context. In the framework, we extract KIs from the reasoning traces of various models and provide these KIs in-context to LLMs when evaluating them. Consequently, gains over a no-KI baseline indicate a knowledge bottleneck, while persistent errors point to reasoning limits.

We first introduce our pipeline to extract KIs from reasoning traces (§4.2), and then discuss how we analyze and apply extracted KIs to test knowledge recall (§4.3, §4.5) and usage (§4.4). For experiments, we prioritize challenging benchmarks (e.g., GPQA, MMLU-Pro*, and LabBench*), which have been widely used by previous work in the field on tasks that require scientific expertise.

**Knowledge Ingredient (KI) Extraction** First, to analyze the role of knowledge in models' performance on scientific problem-solving, we aim to study a setting in which the model is given the requisite knowledge in-context. Specifically, we take the reasoning traces from a reasoning model as the knowledge source and use a strong reasoning LLM (e.g., DeepSeek-R1) to extract the essential atomic knowledge units that comprise it, which we refer to as *knowledge ingredients* (KIs) (Figure 1). A KI is any standalone statement conveying an explicit fact, definition, mechanism, relationship, or insight that can be generalized beyond the specific question. It is a self-contained, generalizable sentence and does not include any contextual or example-specific details.

---

[6]Notably, SYNTHETIC-1-Math is sourced from competition-level math problems, highlighting high-quality abstract math reasoning filtered by verified answers. In contrast, StackExchange and SYNTHETIC-1-STEM provide more realistic problem-solving data from wider subjects, offering more coverage in science domains.

*Table 2.* Performance on GPQA and LabBench* with base models alone, base models with KIs extracted from DeepSeek-R1 or itself (w/ {R1, Base} KIs), and reasoning-fine-tuned models. Best and second-best averages are bold and underlined. Reasoning models fall behind base models augmented with in-context knowledge.

| Setup | GPQA | LabBench* |
|---|---|---|
| Qwen | 35.27 | 32.38 |
| w/ Qwen KIs | 34.24 ± 0.93 | 30.93 ± 1.43 |
| w/ R1 KIs | **47.19 ± 1.53** | **41.40 ± 2.46** |
| Qwen-STEM | 41.63 | 31.75 |
| Qwen-Math | 39.47 | 30.18 |
| Qwen-BOTH | 40.81 | 33.83 |
| General-Reasoner | 35.94 | 35.58 |
| Llama | 28.13 | 33.55 |
| w/ Llama KIs | 29.06 ± 1.44 | 34.40 ± 2.58 |
| w/ R1 KIs | **43.57 ± 0.88** | **42.27 ± 1.60** |
| Llama-STEM | 38.95 | 36.04 |
| Llama-Math | 36.16 | 34.78 |
| Llama-BOTH | 39.43 | 36.61 |
| Llama-Nemotron | 37.95 | 27.78 |

The final prompt we use is carefully tuned based on our manual inspections of the extracted KIs (see Appendix E.1 with examples). We then augment the original question by prepending the extracted set of KIs in-context and ask the models to solve the same problem.

To validate the extraction quality and generalizability, we perform additional checks on DeepSeek-R1, Qwen3-30B-A3B-Thinking-2507, and Gemini-3-Flash as the extractors to ensure that the KIs (a) are task-agnostic (i.e., provide knowledge and facts without referring to specific details in the question or options, e.g., "... as referred to in option B ..."), (b) do *not* leak any part of the final answer, and (c) strictly adhere to the traces as the knowledge source without adding extra information. In manual review, *all* extracted KIs from 300 sampled reasoning traces met these criteria and were consistent with their source reasoning traces. For the following analysis, we use KIs generated by DeepSeek-R1, and provide more details and experiment results with alternative extractors in Appendix E.2.

To prevent the extractor from hallucinating or introducing extraneous facts (i.e., KIs unsupported by the source trace or unnecessary for solving the problem), we feed the generated KIs back to the source model and measure performance. If performance changes materially, this indicates potential leakage of steps or answers. Empirically, we observe no significant change (Table 2, Base vs. w/ Base KIs), suggesting the KIs are answer-agnostic and faithful to the trace. Appendix E.3 further tests this with domain-adjacent KIs mixed from same-subject MMLU-Pro* questions. Further, although it is possible that the knowledge pieces may be irrelevant to the solution (Lyu et al., 2023; Turpin et al., 2023; Lanham et al., 2023), recent high-performing models like DeepSeek-R1 have demonstrated strong reasoning ad-

herence on benchmark tasks (Guo et al., 2025)(we show KI examples generated by different models in Figures 13-15, Appendix E.1). Our experiments show that the knowledge pieces help models on reasoning tasks.

Centered on our main objective of studying knowledge recall and use in reasoning models, we examine the following key research questions: **RQ1:** To what extent can base models benefit from high-quality external knowledge? **RQ2:** To what extent do reasoning-enhanced models benefit from external knowledge? **RQ3:** Does reasoning fine-tuning improve models' ability to surface helpful knowledge?

### 4.3. RQ1: To what extent can base models benefit from high-quality external knowledge?

**Problem Statement.** We investigate the potential improvement from external knowledge by providing KIs to the base models in the prompt when performing scientific reasoning (Figure 1). Here, we focus on two sources for the KIs, which are extracted from their own CoT traces (w/ Base KIs) or from DeepSeek-R1's CoT traces (w/ R1 KIs). To overcome context sensitivity, we report averages and standard deviations across 5 runs with corresponding KIs permuted randomly. We then **investigate whether there are significant gaps between base models augmented with additional KIs in the context, and their corresponding reasoning-fine-tuned models.** To this end, comparisons are made with reasoning-fine-tuned models trained on our controlled data mixtures and with those from concurrent work (i.e., General-Reasoner-7B (Liu et al., 2025) and Llama-Nemotron-Nano-8B (Bercovich et al., 2025)) that use SFT and reinforcement learning based on the same base models.

**Answer to RQ1: As an upper bound, a base model with high-quality in-context knowledge can substantially outperform its reasoning-enhanced counterpart.**

As shown in Table 2, base models provided with KIs from DeepSeek-R1 are able to *outperform* base models alone or Base w/ Base KIs setup by $\geq 20\%$, and *outperform* reasoning variants without KIs by $\geq 10\%$ across different benchmarks and model families, showing the external knowledge provides greater gain than reasoning fine-tuning. The fact that a base model without strong reasoning capabilities can outperform reasoning models in this setting suggests that their parametric knowledge lacks the information in the KIs, or that they struggle to retrieve it from their parametric storage, hindering performance in scientific reasoning.

### 4.4. RQ2: To what extent do reasoning-enhanced models benefit from external knowledge?

**Problem Statement.** Given the large gains from adding DeepSeek-R1 KIs to base models in RQ1, we hypothesize similar improvements would scale on reasoning-enhanced

*Table 3.* Accuracy of Qwen and Llama variants on benchmarks with external knowledge ingredients (KIs). We report averages and standard deviations over 5 random permutations of the KIs. Reasoning variants w/ R1 KIs outperform base model w/ R1 KIs across different benchmarks and models.

| | GPQA | | MMLU-Pro* | | LabBench* | |
|---|---|---|---|---|---|---|
| **Models** | w/ self KIs | w/ R1 KIs | w/ self KIs | w/ R1 KIs | w/ self KIs | w/ R1 KIs |
| Qwen | $34.24 \pm 0.93$ | $47.19 \pm 1.53$ | $59.03 \pm 0.34$ | $68.86 \pm 0.56$ | $30.93 \pm 1.43$ | $41.40 \pm 2.46$ |
| Qwen-STEM | $\mathbf{41.63 \pm 2.10}$ | $52.50 \pm 2.14$ | $64.71 \pm 1.05$ | $69.69 \pm 0.73$ | $31.75 \pm 2.81$ | $43.79 \pm 1.71$ |
| Qwen-Math | $39.47 \pm 1.66$ | $53.53 \pm 1.24$ | $\mathbf{66.93 \pm 0.72}$ | $\mathbf{74.00 \pm 0.59}$ | $30.18 \pm 1.65$ | $41.17 \pm 2.32$ |
| Qwen-BOTH | $40.81 \pm 2.04$ | $\mathbf{54.46 \pm 1.27}$ | $65.71 \pm 0.74$ | $71.64 \pm 1.16$ | $\mathbf{33.83 \pm 2.59}$ | $\mathbf{43.90 \pm 2.71}$ |
| Llama | $29.06 \pm 1.44$ | $43.57 \pm 0.88$ | $47.73 \pm 0.89$ | $60.53 \pm 1.67$ | $34.40 \pm 2.58$ | $42.27 \pm 1.60$ |
| Llama-STEM | $38.95 \pm 1.31$ | $53.17 \pm 1.15$ | $59.14 \pm 0.85$ | $68.19 \pm 1.15$ | $36.04 \pm 3.98$ | $46.87 \pm 1.49$ |
| Llama-Math | $36.16 \pm 2.33$ | $53.75 \pm 1.15$ | $59.65 \pm 0.98$ | $69.01 \pm 0.55$ | $34.78 \pm 4.26$ | $45.55 \pm 0.68$ |
| Llama-BOTH | $\mathbf{39.43 \pm 2.00}$ | $\mathbf{54.73 \pm 1.75}$ | $\mathbf{63.81 \pm 0.90}$ | $\mathbf{72.74 \pm 0.26}$ | $\mathbf{36.61 \pm 2.73}$ | $\mathbf{48.65 \pm 0.49}$ |

*Table 4.* Accuracy (%) of synthetic knowledge recall on KIs generated from Qwen/Llama-Math on GPQA and MMLU-Pro*. Base models and math reasoning-fine-tuned models show similar performance on knowledge recall questions.

| | Qwen | -Math | Llama | -Math |
|---|---|---|---|---|
| KI Dataset | | *Qwen-Math* | | *Llama-Math* |
| KI-GPQA | 72.30 | 73.02 | 70.94 | 68.94 |
| KI-MMLU-Pro* | 82.49 | 81.50 | 74.46 | 74.12 |

*Table 5.* Performance on GPQA and MMLU-Pro* with KIs extracted from base and -Math reasoning models. KIs extracted from -Math models enable more improvement over those from the base.

| Base | Setup | GPQA | MMLU-Pro* |
|---|---|---|---|
| Qwen | w/ Qwen KIs | $34.24 \pm 0.93$ | $59.03 \pm 0.34$ |
| | w/ Qwen-Math KIs | $36.93 \pm 1.75$ | $63.66 \pm 0.45$ |
| Llama | w/ Llama KIs | $29.06 \pm 1.44$ | $47.73 \pm 0.89$ |
| | w/ Llama-Math KIs | $29.69 \pm 1.72$ | $53.91 \pm 0.94$ |

models, offering additional gains on top of enhanced reasoning. To this end, we evaluate base and CoT SFTed variants on KIs extracted from DeepSeek-R1, providing the same necessary knowledge from DeepSeek-R1's reasoning traces (w/ R1 KIs). As a baseline without the added knowledge, we provide the tested models with KIs extracted from their own CoT traces (w/ self KIs) for comparison.

**Answer to RQ2: Reasoning models also substantially benefit from the addition of contextual knowledge.** As shown in Table 3, within both the Qwen and Llama groups, reasoning-enhanced models w/ R1 KIs in the context show significant improvements over the base setting without the KIs, while preserving the gap relative to the base model w/ R1 KIs. Confirming the effectiveness of providing external knowledge as in-context prompts, this result sheds light on potential future improvement by applying high-quality external memory modules as an external knowledge source for better problem-solving capabilities, echoing with COMPACTDB (Lyu et al., 2025), a concurrent effort constructing

a high-quality datastore for reasoning-intensive tasks.

Note, however, that in these experiments, we do not distinguish between two possible non-exclusive explanations for the improvement from adding R1 KIs. (a) It may be that the R1 KIs provide new knowledge absent from the model's parameters, or (b) the model may already possess these facts but struggle to retrieve them (put another way, once a strong reasoning model supplies the *key* facts, the reasoning search space might narrow and the problem becomes easier, whether or not the model originally "knew" the augmented facts). We further analyze this confounder in RQ3, and we include a domain-adjacent KI ablation in Appendix E.3 to stress-test whether the gains are driven by question-specific solution-path hints.

### 4.5. RQ3: Does reasoning fine-tuning improve models' ability to surface helpful knowledge?

**Problem Statement.** While we observe that external knowledge benefits reasoning models, in this RQ, we ask how reasoning-fine-tuning affects knowledge recall. To this end, we focus on evaluating the KIs from -Math models to determine whether they offer more improvement than those of base models, as -Math models are fine-tuned on math-only data without additional scientific knowledge.

Notably, in Table 2, while -STEM and -BOTH variants, trained with SYNTHETIC-1-STEM, outperform -Math variants due to science in-domain training data, -Math variants also largely outperform the base model even without being trained on science data. Recalling our discussion in RQ2 (§4.4), the -Math model's gains have the same two non-exclusive explanations, (a) the -Math model performs better on science questions that require math because math knowledge was loaded into the model through the math-specific fine-tuning, and/or (b) the -Math model is better at surfacing its relevant parametric knowledge via CoT expression.

To disentangle these two factors, we extract KIs from the CoTs of the -Math models and examine whether these KIs

represent new knowledge added by fine-tuning, or whether they are also facts known to the base model. We probe this by querying the model with synthetic questions that test knowledge of each KI (see Appendix E.4 for examples). Then, to verify explanation (b), we provide the KIs in-context from either the -Math or base model, to the corresponding base model; i.e., holding reasoning capacity constant while varying only the external knowledge.

**Answer to RQ3: Yes.** In response to explanation (a), we find that on average, the base models and their corresponding -Math variants have similar recall of the KIs (Table 4), meaning that explanation (a) is unlikely to be the major contributor for the improvements.

To verify explanation (b), Table 5 shows that KIs from -Math deliver significant boosts over those from the base models across different benchmarks and model families. This result suggests that CoT verbalization improves the model's ability to surface relevant knowledge for the given reasoning problems. Notably, the KIs are unlikely to have been newly acquired during fine-tuning (Table 4); instead, the findings indicate that reasoning-fine-tuned models exhibit improved recall of knowledge already parameterized in the base model.

## 5. Conclusion

In this work, we studied how reasoning and domain knowledge each contribute to scientific reasoning in LLMs. To this end, we introduced SCIREAS and SCIREAS-PRO, unified, reproducible suites for evaluating scientific reasoning across domains and formats, together with KRUX, a knowledge-controlled evaluation framework. We showed: (i) retrieving task-relevant knowledge from parameters is a key bottleneck; (ii) reasoning-fine-tuned models get complementary gains from external KIs; and (iii) verbalized CoT improves knowledge surfacing. Our results show that reasoning-fine-tuning improves both reasoning and knowledge use, suggesting future directions in better understanding and enhancing these interconnected components.

## Acknowledgements

This project was supported in part by Google's Research Scholar Program and compute credits from Nvidia through Nvidia's academic grants program. We thank Luca Soldaini and Dirk Groeneveld for helpful discussions in the early stages of the project.

## Impact Statement

This work aims to advance the evaluation and diagnosis of scientific problem-solving in LLMs. By introducing SCIREAS and SCIREAS-PRO, we provide a unified bench-mark suite that consolidates diverse scientific and engineering tasks into a standardized evaluation framework, enabling holistic benchmarking while remaining computationally efficient. By leveraging diverse datasets and question formats, our evaluation suites enable practical assessment of model behavior across multiple benchmarks rather than drawing conclusions from a single task. This holistic view helps surface LLM behaviors that have not been systematically characterized in prior work on scientific problem solving, such as inconsistent model behavior across benchmarks, evidence that different model families appear tuned to different benchmarks, and how performance correlations vary across tasks and domains.

Beyond evaluation, KRUX offers a diagnostic framework that helps model developers jointly optimize knowledge retrieval and reasoning. It provides a controlled way to study whether failures stem from missing knowledge, difficulties retrieving or utilizing knowledge, or limitations in reasoning. This is practically useful for guiding model development decisions, e.g., whether to prioritize better retrieval/grounding, better reasoning post-training, or their combination. Our results also contribute empirical insight into knowledge utilization in reasoning-oriented models, showing that stronger reasoning alone does not guarantee reliable access to relevant scientific knowledge, and that improving knowledge utilization remains an important lever for scientific problem-solving performance.

While advances in scientific reasoning can be misused (e.g., producing persuasive but incorrect scientific explanations), we expect the primary impact of this work to be enabling more systematic, compute-efficient evaluation and more targeted diagnosis, supporting the development of scientific LLMs that are both more capable and more reliable.

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

## A. Extended Related Work

**Evaluating Knowledge of LLMs**    Early efforts tended to evaluate the LM knowledge frontier with a static unified benchmark (Petroni et al., 2021). However, given the growing training corpus for pushing LLM performance, quantifying the knowledge frontier of LLMs becomes increasingly challenging, making it difficult to design a unified benchmark. Instead of general knowledge evaluation, recent work approaches the knowledge frontier of LLMs by anchoring on specific entities, proposing methods to quantify knowledge and factuality around given entities (Gottesman & Geva, 2024; Cohen et al., 2023). With recent development of reasoning LLMs, more work exploits long CoT traces as evidence of explicit knowledge utilization, verifying knowledge recall in CoT traces for factuality (Wu et al., 2025). Nevertheless, directly evaluating CoT traces can result in false positive signals on the knowledge boundary, given that the knowledge involved could be factual but not helpful for problem solving (Arcuschin et al., 2025). In our framework, we construct controlled settings and protocols to evaluate whether the knowledge is genuinely helpful for problem-solving, implicitly guaranteeing the factuality and relevance.

**Reasoning LLMs**    Recent work has shown that LLMs can be trained to utilize intermediate tokens for reasoning, achieving better performance on reasoning tasks as the decoding budget increases. OpenAI's o-series (OpenAI et al., 2024) represents the landmark of this paradigm among commercial frontier models, followed by DeepSeek-R1 (Guo et al., 2025) and several recent efforts to reproduce this success without releasing the training data, such as QwQ (Team, 2025) and Kimi (Team et al., 2025a). Some recent initiatives aim to achieve the same goal using fully open data sources, led by Llama-Nemotron from NVIDIA (Bercovich et al., 2025) and SYNTHETIC-1 from Prime Intellect (Mattern et al., 2025), releasing post-training data to foster development within the community. Our work builds on these commitments, sharing the vision of improving model reasoning by leveraging intermediate tokens, while emphasizing our focus on scientific domains rather than on mathematics or general logical reasoning.

**LLMs for Science**    Recent advancements in scientific LLMs have transitioned from early domain-specific pretraining (e.g., Beltagy et al. 2019; Lee et al. 2019), to more comprehensive models with multiple stages of training, e.g., SciGLM (Zhang et al., 2024), SciLitLLM (Li et al., 2025), and OmniScience (Prabhakar et al., 2025). On the other hand, reasoning models have shown strong performance on scientific tasks such as GPQA and MMLU-Pro (Guo et al., 2025; OpenAI et al., 2024), and some recent efforts instrument LLMs to separate recall from deduction during inference (Wang et al., 2025; Jin et al., 2025). However, we still lack a clear understanding of the factors underlying performance on scientific tasks, such as knowledge acquisition or improved reasoning capabilities. We aim to address this gap by studying these factors and then providing a recipe for training more capable models in science.

## B. SCIREAS Details

*Table 6.* Performance (%) across SCIREAS grouped by models at low and high reasoning efforts. The same model with different reasoning effort can have distinctive performance with a clear margin.

| Benchmark | o3 | | | o3-mini | | | o4-mini | | | Gemini-2.5-Pro | | | Claude-Sonnet-4 | | | GPT-5 | | |
|---|---|---|---|---|---|---|---|---|---|---|---|---|---|---|---|---|---|---|
| | Low | High | Δ | Low | High | Δ | Low | High | Δ | Low | High | Δ | Low | High | Δ | Low | High | Δ |
| GPQA | 75.4 | 79.9 | +4.5 | 63.4 | 73.9 | +10.5 | 69.4 | 74.6 | +5.2 | 80.1 | 79.5 | -0.6 | 63.8 | 69.0 | +5.2 | 79.2 | 82.4 | +3.1 |
| SuperGPQA* | 54.9 | 59.5 | +4.6 | 40.5 | 54.0 | +13.5 | 48.6 | 57.1 | +8.5 | 60.1 | 60.4 | +0.3 | 45.2 | 49.8 | +4.6 | 58.6 | 62.4 | +3.8 |
| MMLU-Pro* | 85.7 | 86.6 | +0.9 | 82.1 | 85.0 | +2.9 | 84.1 | 86.0 | +1.9 | 85.0 | 86.2 | +1.2 | 84.1 | 85.3 | +1.2 | 86.5 | 88.6 | +2.1 |
| LabBench* | 70.5 | 74.2 | +3.7 | 56.9 | 59.2 | +2.3 | 59.7 | 63.7 | +4.0 | 61.9 | 64.4 | +2.5 | 53.4 | 57.2 | +3.8 | 66.6 | 74.4 | +7.8 |
| OlympBench | 53.5 | 58.0 | +4.5 | 39.5 | 51.1 | +11.6 | 40.4 | 49.6 | +9.2 | 67.5 | 69.6 | +2.1 | 55.4 | 59.8 | +4.4 | 60.0 | 64.9 | +4.8 |
| SciBench | 69.7 | 72.1 | +2.4 | 46.0 | 66.3 | +20.3 | 65.5 | 69.7 | +4.2 | 71.0 | 70.2 | -0.8 | 65.5 | 67.1 | +1.6 | 70.4 | 72.0 | +1.6 |
| SciEval* | 84.8 | 82.7 | -2.1 | 83.8 | 83.4 | -0.4 | 87.1 | 87.5 | +0.4 | 86.4 | 85.1 | -1.3 | 85.8 | 85.8 | 0.0 | 87.4 | 86.1 | -1.3 |
| SciKnowEval* | 52.1 | 51.9 | -0.2 | 49.0 | 51.9 | +2.9 | 49.9 | 51.1 | +1.2 | 46.8 | 47.6 | +0.8 | 43.6 | 43.3 | -0.3 | 45.5 | 46.7 | +1.2 |
| SciRIFF* | 51.8 | 53.6 | +1.8 | 51.3 | 51.8 | +0.5 | 50.6 | 52.2 | +1.6 | 51.6 | 51.4 | -0.2 | 53.5 | 50.9 | -2.6 | 46.9 | 50.1 | +3.3 |
| UGPhysics* | 63.1 | 65.2 | +2.1 | 56.7 | 60.7 | +4.0 | 57.7 | 62.2 | +4.5 | 56.0 | 55.4 | -0.6 | 52.4 | 53.2 | +0.8 | 63.6 | 67.6 | +4.0 |
| **Average** | 66.2 | 68.4 | +2.2 | 56.9 | 63.7 | +6.8 | 61.3 | 65.4 | +4.1 | 66.6 | 67.0 | +0.4 | 60.3 | 62.1 | +1.8 | 66.5 | 69.5 | +3.1 |
| **0.01\$ / Instance** | 0.68 | 2.25 | ×3.3 | 0.41 | 3.24 | ×7.9 | 0.41 | 2.38 | ×5.8 | 1.07 | 12.51 | ×11.7 | 1.83 | 7.50 | ×4.1 | 0.72 | 3.10 | ×4.3 |

## B.1. Evaluation Suite Curation

**Subtask Selection Protocol**   Our selection process involves joint decisions from three authors, where the two authors determine targeted tasks together and resolve disagreements by discussion, and another author validates them independently, following the exact selection policy. We outline the main criteria of our selection process below:

1. Most candidate benchmarks we considered have clear documentation in their associated manuscripts that describes the characteristics of each subtask, which are indicative of their difficulty and reasoning intensity. For example, SciKnowEval specifies five difficulty levels for the subtasks: L1: Knowledge Memory; L2: Knowledge Comprehension; L3: Knowledge Comprehension; L4: Knowledge Discernment; L5: Knowledge Application. We relied on these descriptions to determine which subtasks to include. For example, for SciKnowEval we only kept subtasks that are labeled with L3 Knowledge Comprehension and above.

2. We then follow the two-step exclusion policy: for each candidate subtask, we sampled 20 instances and excluded the subtask if any sample failed to both (i) require domain knowledge beyond the prompt and (ii) require multi-step reasoning. This deliberately conservative, exclusion-oriented design looks for reasons to remove subtasks, biasing against inclusion and reducing the risk of false positives (i.e., tasks that are not genuinely reasoning-intensive).

   For example, SciEval evaluates from 4 dimensions: basic knowledge, knowledge application, scientific calculation, and research ability. The basic knowledge subset is excluded in the Step 1 paper inspection phase as it demands limited reasoning capabilities by design. After the exclusion inspection, we only keep the knowledge application and scientific calculation subsets, as the research ability subset contains questions that are research-relevant but not reasoning-intensive.

3. Meanwhile, we also exclude tasks with subjects from niche areas that lie outside mainstream STEM (e.g., weapon science, textile engineering from SuperGPQA).

To validate our filtering results, we have another author independently conduct the filtering process from Step 2 again, following the exact same selection policy and compared with the previous results. Among 111 candidate subtasks/domains, SciReas settled on 75 subtasks (listed in Table 15-16), and the two-fold annotation validation reached an agreement accuracy of 90.1%. The disagreement was mainly due to the scope decision in Step 3 for SuperGPQA where tasks from more domains could be within our scope. For example, fields of "Stomatology" and "Public Health and Preventive Medicine" could be justified as mainstream and contain instances that are scientific reasoning-intensive. For SciReas in our submission, we apply a strict filter and exclude fields like these. We list the selection of each benchmark as follows. See Table 15-16 for domain distribution.

**GPQA (Rein et al., 2024):**   No change. Report in micro average. License: CC-BY-4.0.

**MMLU-Pro (Wang et al., 2024b):**   MMLU-Pro features subjects beyond STEM and scientific subjects. We first filter by subjects, retaining instances from physics, chemistry, computer science, math, biology, and health, and then randomly sample each task to 200 instances max. Report in macro average across 7 subjects. License: MIT.

**LabBench (Laurent et al., 2024):**   We drop tasks that require visual inputs or external table/paper extraction, therefore dropping DbQA, FigQA, LitQA2, SuppQA, and TableQA, retaining CloningScenarios, PropotolQA, and SeqQA. Report in macro average across 3 tasks. License: CC-BY-SA-4.0.

**SciBench (Wang et al., 2024a):**   No change. Report in micro average. License: MIT.

**OlympiadBench (He et al., 2024):**   Dropping tasks that require visual inputs or not in English. Report the macro average across math and physics. License: apache-2.0.

**SciRIFF (Wadden et al., 2025):**   We drop tasks that primarily focus on information/relation/table extraction and retain EvidenceInference, Qasper, and SciFact. Report in macro average of 5 metrics (detailed in Table 15-16) across 3 tasks. License: ODC-BY.

*Figure 6.* Correlation between sampled and full dataset performance as a function of sample size. 200 instances per subject (purple) yields r = 0.919 ± 0.043. Error bars: SD over 30 samples.

*Figure 7.* 95% confidence intervals for 200-instance sampling across nine frontier models. Mean CI half-width ≈ 0.0015; numbers above/below bars show mean and half-width.

**SciKnowEval ([Feng et al., 2025](#)):**   The authors introduce scientific tasks in 5 progressive levels from knowledge memorization to application. After manual inspection, we only preserve tasks from the highest level of knowledge application (L5), and cap instances from each task to be 200. Report the macro average across 8 tasks. License: MIT.

**SciEval ([Sun et al., 2024](#)):**   Similar to SciKnowEval, the authors introduce 4 progressive levels of static tasks, including basic knowledge, knowledge application, scientific calculation, and research creativity. After inspection, we retain knowledge application and scientific calculation subsets, capping each task to a maximum of 200. Report the macro average across 6 tasks. License: N/A.

**UGPhysics ([Xu et al., 2025](#)):**   The authors annotate each instance into 5 different physics reasoning skills: knowledge recall, laws application, math derivation, practical application, and others, which fails to be categorized into the categories above. We filter out instances that specifically require knowledge recall only and cap instances from each subject to be 200 max. Report the macro average across 13 subjects. License: CC-BY-NC-SA-4.0.

**SuperGPQA ([Team et al., 2025b](#)):**   We curate questions from two broad domains — science and engineering — while omitting niche areas that lie outside mainstream STEM (e.g., weapon science, textile engineering). The science portion spans mathematics, biology, physics, systems science, and chemistry. The engineering portion covers a comprehensive set of disciplines: electronic science and technology; nuclear science and technology; mechanical engineering; information and communication engineering; civil engineering; instrument science and technology; computer science and technology; control science and engineering; chemical engineering and technology; mechanics; electrical engineering; materials science and engineering; hydraulic engineering; power engineering and engineering thermophysics; and optical engineering. Report in macro average across the domain of science and engineering. License: ODC-BY.

### B.2. Uniform Sampling Validation: MMLU-Pro Case Study

Evaluating state-of-the-art frontier models could be expensive. To mitigate evaluation cost, we evaluate frontier models on MMLU-Pro* before and after uniform sampling. By sample size correlation in Figure 6 and 95% confidence intervals for sampled subset in Figure 7, we show that the sampling is cost-efficient and statistically effective while reducing evaluating instances from 6,696 to 1,400.

For costly frontier reasoning models such as Gemini-2.5-Pro-Preview, at rates in time of writing, the sampling reduces SCIREAS evaluation costs from $3,600 to $1,500 and can be further decreased to $730 by using batch job inference.

## B.3. SCIREAS-PRO Reasoning Intensiveness Validation

To test this hypothesis, we pursue two complementary checks: (1) different reasoning models should have high agreement identifying reasoning intensive instances, and (2) filtered instances should agree with human judgment in terms of reasoning intensiveness.

### B.3.1. CROSS-MODEL AGREEMENT ON REASONING INTENSITY

To validate our hypothesis that performance gaps between different reasoning effort settings indicate reasoning intensity, we first examine whether different models agree on which instances are reasoning-intensive. As shown in Figure 1, for each reasoning model, we categorize each test question from SCIREAS by their correctness under low/high reasoning efforts into four categories, (high_c, low_c), (high_c, low_i), (high_i, low_c), and (high_i, low_i), where high/low stands for high/low reasoning effort setting and *_c/*_i stands for the problem instance has been answered correctly/incorrectly by the model. Treating (high_c, low_i) as targeting instances that require high reasoning effort, we measure how (high_c, low_i) sets derived from different reasoning models agree with others.

As shown in Table 7, treating (high_c, low_i) from o3-mini as ground truth, the same set derived from o4-mini, o3, and claude-sonnet-4 largely coincide with o3-mini across different benchmarks from SCIREAS (all above 70%), showing high agreement on instances that require high reasoning efforts across models from different model families.

*Table 7.* Accuracy of overlapping instances on (high_c, low_i) from o3-mini vs. other models, treating o3-mini as ground true label. Different reasoning models agree on high reasoning instances.

| Ground Truth

Target | o3-mini
vs.
o4-mini | o3-mini
vs.
o3 | o3-mini
vs.
claude-sonnet-4 |
|---|---|---|---|
| SuperGPQA* | 78.0 | 77.8 | 76.1 |
| GPQA | 80.4 | 81.0 | 79.0 |
| MMLU-Pro* | 92.2 | 91.6 | 92.0 |
| LabBench* | 71.9 | 74.6 | 75.8 |
| SciBench | 75.9 | 74.1 | 75.4 |
| OlympiadBench | 81.1 | 81.5 | 81.2 |
| SciEval* | 94.3 | 93.1 | 93.5 |
| UGPhysics* | 83.2 | 82.9 | 83.8 |

### B.3.2. OPEN-SOURCE SELECTOR VALIDATION

To test whether SCIREAS-PRO construction depends on proprietary selectors, we construct SCIREAS-PRO-Qwen by selecting instances that Qwen3-32B answers correctly with thinking mode on and incorrectly with thinking mode off. This produces 3,208 examples. As shown in Table 8, SCIREAS-PRO-Qwen amplifies the performance gaps between low- and high-reasoning settings across open and closed model families, suggesting that the selection procedure is not specific to proprietary OpenAI models.

*Table 8.* Performance (%) on SCIREAS and SCIREAS-PRO-Qwen. The open-source-selector variant preserves the amplified gap between low- and high-reasoning settings.

| | o3 | | Claude-Sonnet-4 | | DeepSeek-V3/R1 | | GPT-OSS | |
|---|---|---|---|---|---|---|---|---|
| Level | SCIREAS | SCIREAS-PRO-Qwen | SCIREAS | SCIREAS-PRO-Qwen | SCIREAS | SCIREAS-PRO-Qwen | SCIREAS | SCIREAS-PRO-Qwen |
| Low | 66.2 | 73.4 | 60.3 | 55.1 | 51.7 | 40.3 | 59.1 | 54.3 |
| High | 68.4 | 77.8 | 62.1 | 62.3 | 64.2 | 73.6 | 61.3 | 61.0 |
| Gap | +2.2 | +4.4 | +1.8 | +7.2 | +12.5 | +33.3 | +2.2 | +6.7 |

### B.3.3. HUMAN AND LLM-AS-JUDGE ASSESSMENT

The overlap of instances that require high reasoning effort shows reasoning models tend to agree on problem difficulty, but to verify the reliability of reasoning effort as a surrogate, the filter should also align with human judgment.

To this end, we collect the union of (high_c, low_i) from o3-mini and o4-mini for the case study and apply an LLM-as-judge assessment (Zheng et al., 2023) to expedite the process while manually annotating a subset for a reliability test. The LLM judge is based on GPT-4.1 for a balanced tradeoff between assessment reliability and cost. Notably, naively prompting the LLM judge to determine the reasoning difficulty could be suboptimal due to a lack of reference. Therefore, we designed two reference-based evaluation protocols: (a) pair-wise comparison on reasoning difficulty between instance questions sampled from filtered subset and original SCIREAS, and (b) identifying failing reason for filtered instances given low and high reasoning outputs (i.e., whether the model fails in a low reasoning setting due to lack of reasoning effort).

**(a) Pairwise Comparison** For each instance in SCIREAS-PRO, the judge is also presented with an instance drawn from the set of other, non-overlapping instances from SCIREAS. The judge is not given any information as to which instance is

drawn from which source and is tasked to identify which instance is more reasoning-intensive.

---

**SYSTEM MESSAGE**

You are an expert judge comparing reasoning intensity between two questions. Analyze both questions thoroughly and determine which one demands more complex reasoning.
Reply in this exact format:
```
###EXPLANATION: <detailed analysis of both questions and the comparison>
###RESULTS: A / B / UNCLEAR
```

---

**USER MESSAGE**

You will be shown two questions (A and B) from the same academic domain.
A question is *reasoning intensive* if it requires:
• Complex multi-step logical reasoning
• Advanced mathematical computation or derivation
• Integration of multiple concepts or principles
• Abstract thinking or sophisticated problem-solving strategies
• Deep domain knowledge application
*QUESTION A*
Context: {{context_a}}
Question: {{question_a}}
*QUESTION B*
Context: {{context_b}}
Question: {{question_b}}
Analyze both questions carefully and explain your reasoning. Then reply using the exact format specified above.

---

*Figure 8.* Full reasoning intensiveness pairwise comparison prompt template used in our experiments.

**(b) Failure Analysis**  For each instance in SCIREAS-PRO, the judge is presented with both the correct high reasoning output (if both o3-mini-high and o4-mini-high are correct, o4-mini-high will be selected) as well as the incorrect low reasoning output from the corresponding model (e.g. correct: o3-mini-high; incorrect: o3-mini-low). The judge is tasked with determining whether the failure of the low reasoning effort model can be attributed primarily due to insufficient reasoning ability or lack of domain knowledge.

---

**SYSTEM MESSAGE**

You are an expert judge analyzing why AI models fail on reasoning-intensive questions. Compare the correct and incorrect answers to determine if the failure was primarily due to insufficient reasoning ability or lack of domain knowledge.
Reply in this exact format:
```
###EXPLANATION: <detailed analysis of why the low-reasoning model failed>
###RESULTS: REASONING/KNOWLEDGE/BOTH/UNCLEAR
```

---

**USER MESSAGE**

You will be shown a question from an academic dataset, along with
(1) a *CORRECT* answer from a high-reasoning model and
(2) an *INCORRECT* answer from a low-reasoning model.
Your task is to analyze *why* the low-reasoning model failed.
Consider whether the failure is primarily due to:
- *REASONING*: Insufficient logical thinking, problem-solving ability, or step-by-step analysis
- *KNOWLEDGE*: Lack of domain knowledge (missing facts, formulas, concepts, procedures)
- *BOTH*: Significant deficiencies in both reasoning and knowledge
- *UNCLEAR*: Cannot determine the primary cause of failure
QUESTION
Context: {{context}}
Question: {{question}}
CORRECT ANSWER (from {{high_model}}):
{{high_full_response}}
INCORRECT ANSWER
(from {{low_model}}):
{{low_full_response}}
Analyze why the low-reasoning model failed. Was it primarily due to insufficient reasoning ability or lack of knowledge?

*Figure 9.* Prompt used to classify failure cause (reasoning vs. knowledge) for low-reasoning models.

**Results**   We show that both protocols agree that filtered instances require significantly more reasoning efforts than non-filtered instances from SCIREAS, with (a) showing 71% agreement in accuracy by LLMs with 78% human annotation agreement and (b) showing 91% agreement by LLMs with 90% human agreement, where human annotations are made by authors on 80 sampled tests for each protocol.

## C. Frontier Model API Evaluation Configuration

For OpenAI and xAI provided reasoning models, we apply generic "low" and "high" reasoning effort parameters with respect to official documentation where specificity on token budget is not allowed; for other reasoning models that allows thinking budgets as input (e.g. Gemini and Anthropic), we adopt "low" as definition introduced by LiteLLM,[7] which corresponds to 1024 budget, and remove the constraint to allow for as many thinking tokens as the model needed to unleash full potential as "high" reasoning effort, corresponding to the highest reasoning effort from OpenAI and xAI models. For all frontier reasoning models, if not restricted, we set temperature=1, borrowed from OpenAI forced setting,[8] and top-p=0.95, borrowed from recommended setting by Anthropic,[9] with max generation length of 64K, as we observe no models tend to output more than 20K tokens. We log API pricing at the time of writing in Table 9.

---

[7]https://docs.litellm.ai/docs/providers/anthropic#usage—thinking–reasoning_content
[8]https://community.openai.com/t/o3-mini-unsupported-parameter-temperature/1140846/3
[9]https://docs.anthropic.com/en/docs/build-with-claude/extended-thinking#feature-compatibility

*Table 9.* Pricing ($ per 1M tokens) for input and output across different LLM providers at the time of writing, without any discounts.

| Model | Input Price ($ per 1M tokens) | Output Price ($ per 1M tokens) |
|---|---|---|
| *OpenAI models* | | |
| GPT-4.1-2025-04-14 | 2.00 | 8.00 |
| o3-mini-2025-01-31 | 1.10 | 4.40 |
| o3-2025-04-16 | 2.00 | 8.00 |
| o4-mini-2025-04-16 | 1.10 | 4.40 |
| GPT-5-2025-08-07 | 1.25 | 10.00 |
| GPT-oss-120B (Together AI) | 0.15 | 0.60 |
| *DeepSeek models* | | |
| DeepSeek-V3-0324 | 0.14 | 0.28 |
| DeepSeek-R1-0120 | 0.55 | 2.19 |
| DeepSeek-R1-0528 | 0.55 | 2.19 |
| *Alibaba Qwen models (Together AI)* | | |
| Qwen3-32B | 0.40 | 1.20 |
| Qwen3-235B-2507 | 0.65 | 3.00 |
| *Google models* | | |
| Gemini-2.5-Pro-Preview-05-06 | 1.25 | 10.00 |
| *Meta models (Together AI)* | | |
| Llama-4-Maverick-17B-128E-Instruct-FP8 | 0.27 | 0.85 |
| *Anthropic models* | | |
| Claude-Sonnet-4-20250514 | 3.00 | 15.00 |

# D. Training / Evaluation Details

## D.1. Distillation from Reasoning LLMs

To obtain high-performing reasoning models for study, we employ a distillation method that fine-tunes smaller models using Supervised Fine-tuning (SFT) on the CoT trajectories generated by large reasoning models, as it is more effective than reinforcement learning (RL) with the small models alone (Guo et al., 2025). Specifically, we consider the standard SFT framework for language models where the objective is to train a model $f_\theta$ to approximate a distribution over output sequences $y$ conditioned on input $x$, based on a dataset $\mathcal{D} = \{(x^i, y^i)\}_{i=1}^N$. For recent reasoning LLMs such as DeepSeek-R1, the output $y$ consists of two main parts: a reasoning trace $r$ and the actual output $a$. In practice, the reasoning traces are enclosed by keywords `<think>` and `</think>`, indicating the start and the end of the reasoning process. The model is trained with the standard SFT objective: $\mathcal{L}(\theta) = -\Sigma_{(x,y)\in\mathcal{D}}\Sigma_{t=1}^{|y|} \log p_\theta(y_t|y_{<t}, x)$, where $y_t$ is the $t$-th token and $y_{<t}$ is its prefix.

## D.2. Extended Setup

### D.2.1. TRAINING SETTINGS

We filter out instances with a token length greater than 4096.[10] The models are trained for 5 epochs with a cosine learning rate scheduler, a maximum learning rate of 1e-5, and 3% warmup steps.

### D.2.2. EVALUATION SETUP

The reasoning models could produce excessively long outputs, and may be prone to self-repetition with greedy decoding (Guo et al., 2025). In this work, unless otherwise specified, we apply greedy decoding on non-CoT fine-tuned models and top-p=0.95, temperature=0.6 on reasoning models, with a maximum generation length of 64K. From our preliminary studies, we observe that the setup generally reflects the best performance for both settings, and the decoding setup matches the recommended setup from recent efforts in large reasoning models, such as Llama-Nemotron (Bercovich et al., 2025). Notably, for Qwen (Qwen et al., 2025) models and their variants, we apply YaRN context extension (Peng et al., 2023) as recommended by the official model card (Qwen et al., 2025).

---

[10]Longer input lengths would slow down our training in quadratic order based on 8 80GB A100/H100 GPUs.

**Units recognised by the heuristic**

- % °C, °F, K, °

- g, kg, mg, $\mu g$/ug, lb/lbs, oz

- m, cm, mm, km L/l, mL/ml, $\mu L$/$\mu l$/ul

- Pa, kPa, MPa, atm, bar, mbar

- J, kJ, MJ; W, kW, MW, GW

- V, kV; A, mA, $\mu A$/uA

- Hz, kHz, MHz, GHz

- $cm^2$, $m^2$, $mm^2$, $km^2$

- $cm^3$, $m^3$, $mm^3$, $km^3$

- mol; M, mM, $\mu M$/uM, nM, pM

- dB; rpm; rad/s

- s, ms, $\mu s$/us, ns; min, h day/days; yr/yrs

*Figure 11.* Unit suffixes accepted by the numeric heuristic. A standalone number with any of these units (or no unit) is treated as evidence that the question contains mathematical content.

*Table 10.* Accuracy breakdown on math and non-math instances for SCIREAS-PRO. -Math and -STEM variants contribute to different dimensions of performance, while -BOTH captures improvements on both.

| Model | Has-Math Acc. | No-Math Acc. |
|---|---|---|
| SCIREAS-PRO: *1,260 Instances* | | |
| # | 1,172 | 88 |
| Qwen | 14.25 | 12.50 |
| Qwen-STEM | 15.53 | 23.86 |
| Qwen-Math | 17.58 | 13.64 |
| Qwen-BOTH | 20.56 | 28.41 |
| Llama | 11.52 | 13.64 |
| Llama-STEM | 14.16 | 15.91 |
| Llama-Math | 17.24 | 13.64 |
| Llama-BOTH | 15.96 | 23.86 |

## D.3. Math vs. Non-Math

### D.3.1. FILTERING HEURISTICS

We label instances as math-needed if they contain explicit numeric quantities that typically imply computation. Importantly, numbers that appear solely within unit expressions (e.g., "$cm^2$") or chemical formulas (e.g., "$H_2O$" or "NaCl") are not treated as indicators of math-related reasoning.

Specifically, a question is marked *Has-Math* when it includes

1. a signed or unsigned integer/decimal (e.g. `3`, `-2.5`, `60`, `9.81`),
2. **not** embedded inside a word (so digits in `H2O`, `COVID-19`, `IL-2`... are ignored), and
3. optionally followed—without intervening letters—by *any one* of the unit strings listed in Fig. 11.

### D.3.2. COT IMPROVEMENTS ON MATH VS. NON-MATH

In response to the question in §3.2, we use categorize instances from SCIREAS into Has-Math and No-Math, resulting in 8,527 cases identified as Has-Math and 4,757 as No-Math. We compute the micro accuracy on frontier models and plot the performance gains by increasing the thinking budget from low reasoning effort to high reasoning effort in Figure 10.

### D.3.3. EFFECTS ON REASONING-FINE-TUNED MODELS

As shown in Table 1, Qwen-STEM and Qwen-Math both exhibit significant improvement over the base model on SCIREAS and SCIREAS-PRO. Qwen-Math slightly outperforms Qwen-STEM on SCIREAS and the gap is amplified on SCIREAS-PRO.

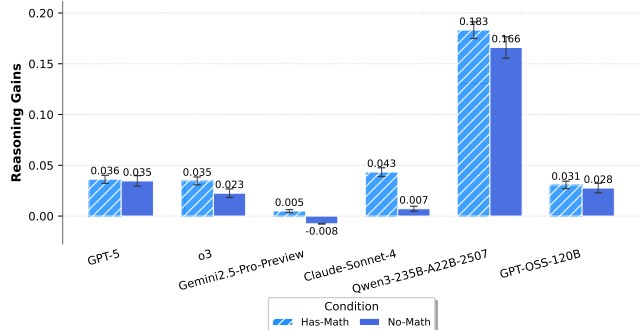

*Figure 10.* Performance gains on Has-Math instances vs. No-Math instances across different frontier models when reasoning effort increases. CI95% shown as the error bar. CoT helps more with Has-Math instances.

Given limited subject coverage on SYNTHETIC-1-Math dataset, the strong performance of checkpoints fine-tuned on it only seems surprising — Does the improvement come from generalization from math reasoning to a wider domain, or is it because the high-reasoning instances in our datasets are math-intensive? To answer this question, we categorize SCIREAS-PRO into math and non-math instances by heuristics.

As shown in Table 10, we find that math computation appears frequently among reasoning-intensive instances, and the improvements on SCIREAS-PRO mostly come from improved math capabilities. For non-math instances, -math variants hardly improve, while -STEM variants and -BOTH variants, trained with STEM subjects data, show noticeable improvements.

## D.4. Training Knowledge Enhanced Scientific Reasoning Models

Our post-trained checkpoints are based on models fine-tuned on either SYNTHETIC-1-Math, SYNTHETIC-1-STEM, or both, while combining the two, which cover both STEM and mathematical reasoning, achieves the strongest performance (Table 1). To further assess the effectiveness of this Math+STEM data mixture following §4.1, we compare it directly against concurrently released long-CoT SFT datasets on the same base model. We then apply the same mixture to Qwen3-8B-Base to obtain SCILIT01 to provide a stronger baseline.

Specifically, we compare Qwen-BOTH, which is fine-tuned using our training recipe, with SYNTHETIC-1-SFT (Mattern et al., 2025), a model fine-tuned on SYNTHETIC-1 with additional coding and preference alignment data, and Qwen-Nemotron, a model we trained with the same settings and same amount of data (§4.1) sampled from science and math domains of Llama-Nemotron (Bercovich et al., 2025), a training data mixture for reasoning fine-tuning, all post-trained on Qwen2.5-7B-Instruct. The results in Table 11 show that our data composition yields a stronger baseline for scientific reasoning than concurrent data recipes on Qwen2.5-7B-Instruct (Table 11 center block), and Qwen-BOTH reaches comparable performance to models from concurrent efforts focusing on reasoning enhancement post-training recipes (Table 11 left-hand block, i.e., OpenR1 (Hugging Face, 2025), Llama-Nemotron (Bercovich et al., 2025), and General-Reasoner (Ma et al., 2025b)).

Furthermore, using our recipe, we fine-tune the recently released Qwen3-8B-Base to deliver a stronger model, SCILIT01. While its performance falls behind Qwen3-8B with the thinking mode, which has undergone more sophisticated post-training, it outperforms Qwen3-8B with non-thinking mode (Table 11 right-hand block). This indicates that SCILIT01 partially unleashes the reasoning capabilities from the base model, offering a strong baseline for future study on post-training recipe for scientific reasoning.

*Table 11.* Performance of concurrent efforts on open-recipe post-training in <10B-parameter level. SCILIT01 shows competitive performance relative to concurrent reasoning post-training methods. We abbreviate Qwen2.5, Qwen3, and Llama-3.1 as Q2.5, Q3, and L3.1, respectively; '-Inst.' denotes the instruction-tuned variant. The best and second-best overall results are highlighted in bold and underlined, respectively.

| Models | OpenR1 | Llama-Nemotron | General-Reasoner | SYNTHETIC-1-SFT | Qwen-Nemotron | Qwen-BOTH | SCILIT01 | Qwen3 | Qwen3-thinking |
|---|---|---|---|---|---|---|---|---|---|
| Base Model | Q2.5-Math | L3.1-Inst. | Q2.5-Base | Q2.5-Inst. | | | Q3-Base | | |
| Training Methods | SFT | SFT&RL | RL | SFT | SFT | SFT | SFT | – | – |
| Trained by Us | No | No | No | No | Yes | Yes | Yes | No | No |
| GPQA | 44.42 | 37.95 | 35.94 | 38.84 | 44.20 | 40.63 | 50.89 | 55.80 | 55.80 |
| SuperGPQA* | 31.90 | 29.39 | 14.26 | 22.39 | 19.47 | 20.33 | 30.11 | 23.32 | 38.27 |
| MMLU-Pro* | 60.86 | 65.64 | 62.14 | 56.21 | 63.57 | 65.00 | 76.57 | 73.36 | 81.71 |
| LabBench* | 27.14 | 27.78 | 35.58 | 28.61 | 35.76 | 33.00 | 35.07 | 36.99 | 38.19 |
| OlympiadBench | 53.03 | 37.62 | 19.82 | 40.75 | 29.33 | 34.55 | 43.78 | 28.51 | 21.30 |
| SciBench | 61.85 | 57.66 | 19.08 | 51.59 | 48.27 | 47.11 | 61.27 | 54.05 | 68.21 |
| SciEval* | 43.64 | 68.67 | 70.34 | 46.41 | 38.53 | 72.36 | 80.60 | 81.51 | 84.02 |
| SciKnowEval* | 28.45 | 30.69 | 34.19 | 19.13 | 31.85 | 32.00 | 39.46 | 37.99 | 41.81 |
| SciRIFF* | 29.17 | 34.01 | 37.75 | 28.57 | 39.24 | 41.81 | 44.01 | 47.23 | 47.26 |
| UGPhysics* | 50.30 | 45.92 | 20.86 | 43.96 | 46.52 | 40.03 | 52.28 | 30.98 | 59.81 |
| **Average** | 43.08 | 43.53 | 34.99 | 37.64 | 39.67 | 42.68 | 51.41 | 46.97 | **53.64** |
| **SCIREAS-PRO** | 26.43 | 23.75 | 13.73 | 19.44 | 19.68 | 21.11 | 24.84 | 19.05 | **29.92** |

# E. Extended KRUX Details

## E.1. Knowledge Extraction

In this work, we apply DeepSeek-R1 as the extractor. Prompt shown in Figure 12. We show a set of KIs extracted from Qwen2.5-7B-Instruct (Figure 13), Qwen-Math variants (Figure 14), and DeepSeek-R1 (Figure 15) for the same question from GPQA:

```
Question: A large gene has dozens of exons, of which the central ones code for
folded triple helical repeats that connect the cytoskeleton with sarcolemma and
extracellular space. Each exon usually codes for one folded triple alpha helix.
The most common mutations of the gene are central exon deletions that create
out-of-frame peptides and progressive degenerative organ waste. A solution
is to deliver a Morpholino that recognizes the 5' end of the out-of-frame exon
in pre-mRNA. The molecule prevents binding of the spliceosome and creates exon
skipping and in-frame joining. Several missing exons are well tolerated by an
organism. Which structure below is not involved in the proposed therapy? (A)
lariat (B) antisense (C) R-loops (D) polyA tail.
```

---

**USER MESSAGE**

You are given a reasoning chain that explains and justifies a particular conclusion or answer. Your task is to extract **all distinct knowledge pieces** from this chain. A knowledge piece is any standalone statement conveying an explicit fact, definition, mechanism, relationship, or insight that can be generalized beyond the specific question.
## Instructions:

1. Read the entire reasoning chain.

2. Identify each discrete fact or insight expressed.

3. Rewrite each as a self-contained, generalizable sentence.

4. Do **not** include any contextual or example-specific details.

5. Output **only** a list of those sentences.

## Output Format:
• knowledge-Piece-1
• knowledge-Piece-2
• ...
## Reasoning Chain: {{REASONING}}
## Now perform the extraction.

---

*Figure 12.* Full knowledge extraction prompt template used in our experiments.

---

**KNOWLEDGE EXTRACTION OUTPUT**

Based on the reasoning chain, here are all distinct knowledge pieces:
1. R-loops are structures formed when a strand of RNA hybridizes with its complementary DNA strand, displacing the other DNA strand.
2. Morpholino can prevent spliceosome binding.
3. Preventing spliceosome binding can lead to exon skipping.
4. Exon skipping can result in in-frame joining of exons.
5. The process of using Morpholino to induce exon skipping does not involve R-loops.

---

*Figure 13.* Example KIs extracted from Qwen2.5-7B-Instruct.

---

**KNOWLEDGE EXTRACTION OUTPUT**

Based on the reasoning chain, here are all distinct knowledge pieces:
1. Morpholino oligonucleotides can bind to the 5' splice site of pre-mRNA to block spliceosome recognition.
2. Blocking spliceosome binding prevents normal intron removal and leads to exon skipping.
3. Exon skipping can enable in-frame joining of adjacent exons, preserving the reading frame.
4. Antisense molecules function by complementary base-pairing to specific RNA sequences to modulate splicing.
5. Lariat structures are formed during canonical splicing via branch site interactions but are absent when spliceosome activity is inhibited.
6. R-loops are DNA-RNA hybrid structures unrelated to spliceosome-mediated exon skipping mechanisms.
7. The polyA tail is added post-transcriptionally and is not directly involved in splice site selection or exon skipping processes.

---

*Figure 14.* Example KIs extracted from Qwen-Math.

---

**KNOWLEDGE EXTRACTION OUTPUT**

Based on the reasoning chain, here are all distinct knowledge pieces:
1. Morpholinos are antisense oligonucleotides that bind to complementary RNA sequences.
2. The spliceosome recognizes splice sites on pre-mRNA through components like U1 snRNP, which binds to the 5' splice site.
3. Blocking spliceosome access to a splice site prevents recognition of exon boundaries and causes exon skipping.
4. Exon skipping can restore the reading frame by joining exons flanking the skipped exon.
5. Functional proteins may be produced if skipped exons encode non-critical regions and the remaining exons maintain an in-frame sequence.
6. Lariat structures form during canonical splicing when introns are removed via branch site cleavage.
7. R-loops are nucleic acid structures involving RNA-DNA hybrids and are not directly involved in spliceosome-mediated splicing.
8. Antisense oligonucleotide therapies rely on sequence-specific binding to pre-mRNA targets.
9. Polyadenylation (polyA tail addition) occurs during mRNA maturation and is unrelated to splice site selection or exon skipping mechanisms.

---

*Figure 15.* Example KIs extracted from DeepSeek-R1.

## E.2. Alternative Extractor

### E.2.1. QUALITY INSPECTION

We generate KIs from the same set of knowledge traces as the knowledge source and inspect them in pairs blindly. Specifically, we compare Qwen3-30B-A3B-Thinking-2507, Gemini-3-Flash, and DeepSeek-R1 as extractors. Across 300 sampled reasoning traces, *all* inspected KIs generated from the three extractors (a) do not refer to any specific identities in the question, e.g., "The option B ...", (b) do not expose the final answers, and (c) adhere to the given traces with no additional information nor missing essential components.

However, different extractors may provide KIs at different granularities, where one could contain more details than another. For example, given the same CoT traces on a math problem, DeepSeek-R1 and Qwen3 generate KIs as shown in Figure 16. Notably, the two sets of KIs **cover nearly identical problem-specific knowledge**, although DeepSeek phrases its conversion rule abstractly (in terms of quantities, rather than Qwen's problem-specific cups and gallons) and Qwen3 includes a problem-solving heuristic to avoid common errors.

### E.2.2. USING ALTERNATIVE EXTRACTOR ON KRUX

Running KRUX with KIs extracted by Qwen3-30B-A3B-Thinking-2507 (Qwen3 for short), Gemini-3-Flash, and DeepSeek-R1, we present Table 12-13 to show that the trends are robust to the extractor choice. On GPQA, Table 12 includes all three extractors; on MMLU-Pro*, Table 13 compares Qwen3 and DeepSeek-R1. With alternative extractors, setups with DeepSeek-R1's KIs in-context (w/ R1 KIs) show significant improvements over the base setups with KIs extracted from the model itself (w/ self KIs). Notably, performance improvements persist in both base models and reasoning-enhanced models, demonstrating that KRUX generalizes to different extractor models.

---

**KI EXAMPLE FROM DeepSeek-R1**

1. One gallon is equivalent to 16 cups.
2. Doubling a quantity requires multiplying the original amount by two.
3. To convert a quantity from a smaller unit to a larger unit, divide the quantity by the number of smaller units per larger unit.

---

**KI EXAMPLE FROM Qwen3-30B-A3B-Thinking-2507**

1. Doubling an amount means multiplying it by two.
2. The number of cups in a gallon is 16.
3. To convert cups to gallons, divide the number of cups by the number of cups per gallon.
4. The correct answer must be in the units requested by the question.
5. A common error is to provide the answer in the intermediate unit rather than the required final unit.

---

*Figure 16.* An example of KIs generated by DeepSeek-R1 and Qwen3-30B-A3B-Thinking-2507. The DeepSeek-R1 tends to cover knowledge recalled in more detail.

*Table 12.* Accuracy of Qwen and Llama variants on GPQA with external knowledge ingredients (KIs) extracted by Qwen3-30B-A3B-Thinking-2507, DeepSeek-R1, or Gemini-3-Flash as the extractor. w/ R1 KIs setups outperform w/ self KIs setups across different models.

| Extractor Setup | Qwen3 | | | DeepSeek-R1 | | | Gemini-3-Flash | | |
|---|---|---|---|---|---|---|---|---|---|
| | w/ self KIs | w/ R1 KIs | Δ | w/ self KIs | w/ R1 KIs | Δ | w/ self KIs | w/ R1 KIs | Δ |
| Qwen | 35.0 | 43.5 | +8.5 | 34.2 | 47.2 | +13.0 | 35.0 | 46.7 | +11.7 |
| Qwen-STEM | 42.6 | 51.1 | +8.5 | 41.6 | 52.5 | +10.9 | 42.2 | 54.0 | +11.8 |
| Qwen-MATH | 38.8 | 50.2 | +11.4 | 39.5 | 53.5 | +14.1 | 40.6 | 51.6 | +11.0 |
| Qwen-BOTH | 43.1 | 52.9 | +9.8 | 40.8 | 54.5 | +13.7 | 44.9 | 52.2 | +7.3 |
| Llama | 29.0 | 39.5 | +10.5 | 29.1 | 43.6 | +14.5 | 29.5 | 42.0 | +12.5 |
| Llama-STEM | 38.6 | 51.8 | +13.2 | 39.0 | 53.2 | +14.2 | 40.8 | 55.4 | +14.6 |
| Llama-MATH | 33.9 | 50.0 | +16.1 | 36.2 | 53.8 | +17.6 | 39.1 | 54.9 | +15.8 |
| Llama-BOTH | 39.1 | 52.2 | +13.2 | 39.4 | 54.7 | +15.3 | 43.3 | 56.7 | +13.4 |

### E.3. Domain-Adjacent KI Ablation

To further test whether KI gains come from answer leakage or question-specific solution-path hints, we create a domain-adjacent KI setting on MMLU-Pro*. For each question, we mix the matched KIs with KIs sampled from five other questions in the same MMLU-Pro subject, then randomly permute the KIs before injection. If KIs were primarily encoding intermediate reasoning steps for a specific question, the additional adjacent KIs should substantially reduce or eliminate the gains. Instead, Table 14 shows consistent improvements over the self-KI baseline for both base and reasoning-enhanced models, supporting the interpretation that KRUX-derived KIs provide generalizable domain knowledge rather than answer-specific reasoning traces.

### E.4. Knowledge Probing

We provide our probing question synthesis prompt (Figure 17), example input and output (Figure 18), and knowledge probing results in Table 4.

*Table 13.* Accuracy of Qwen and Llama variants on MMLU-Pro* with external knowledge ingredients (KIs) extracted by Qwen3-30B-A3B-Thinking-2507 or DeepSeek-R1 as the extractor. w/ R1 KIs setups outperform w/ self KIs setups across different models.

| Extractor Setup | Qwen3 | | | DeepSeek-R1 | | |
|---|---|---|---|---|---|---|
| | w/ self KIs | w/ R1 KIs | Δ | w/ self KIs | w/ R1 KIs | Δ |
| Qwen | 61.7 | 66.4 | +4.6 | 59.0 | 68.9 | +9.8 |
| Qwen-STEM | 63.8 | 69.9 | +6.1 | 64.7 | 69.7 | +5.0 |
| Qwen-MATH | 65.2 | 73.6 | +8.4 | 66.9 | 74.0 | +7.1 |
| Qwen-BOTH | 65.4 | 71.4 | +6.0 | 65.7 | 71.6 | +5.9 |
| Llama | 49.0 | 56.4 | +7.4 | 47.7 | 60.5 | +12.8 |
| Llama-STEM | 58.8 | 65.7 | +6.9 | 59.1 | 68.2 | +9.1 |
| Llama-MATH | 60.7 | 66.4 | +5.7 | 59.7 | 69.0 | +9.4 |
| Llama-BOTH | 62.6 | 70.2 | +7.6 | 63.8 | 72.7 | +8.9 |

*Table 14.* Accuracy (%) on MMLU-Pro* with domain-adjacent KIs. Each prompt mixes the question's matched KIs with KIs from five same-subject questions and randomly permutes them.

| Setup | w/ self KIs | w/ adjacent KIs | Δ |
|---|---|---|---|
| Qwen | 59.0 | 64.9 | +5.9 |
| Qwen-STEM | 64.7 | 69.3 | +4.6 |
| Qwen-Math | 66.9 | 71.6 | +4.7 |
| Qwen-BOTH | 65.7 | 71.0 | +5.3 |
| Llama | 47.7 | 58.1 | +10.4 |
| Llama-STEM | 59.1 | 64.3 | +5.2 |
| Llama-Math | 59.7 | 63.3 | +3.6 |
| Llama-BOTH | 63.8 | 69.1 | +5.3 |

---

**USER MESSAGE**

You are a meticulous question-authoring assistant. Your job is to convert declarative knowledge statements into *probing* multiple-choice questions that can test whether another language model truly stores the fact in its parametric memory.
## IMPORTANT INSTRUCTIONS FOR QUESTIONS:
1. Factual: It should be about a specific detail or fact mentioned in the statement. For example, a true or false statement, a statistic, a definition, etc.
2. Important: It should be a question about the main topic or a key detail/finding/conclusion of the statement.
3. Context-Independent: It should be fully understandable on its own, without phrases like "the proposed model" or "this approach" that assume prior context.
## IMPORTANT INSTRUCTIONS FOR ANSWERS:
1. Provide one correct answer and 4 - 6 incorrect answers.
2. Ensure all answers are roughly the same length and follow a similar style so the correct answer cannot be guessed based on length or style alone.
3. The incorrect answers must be plausible but ultimately wrong, reflecting a misunderstanding or misinterpretation of the knowledge.
## OUTPUT FORMAT: Please provide the question, correct answer, incorrect answers, and a list of text snippets from the article as "evidences" in the following format:
{ "question": "Your question here",
"correct_answer": "Correct answer here",
"incorrect_answers": ["Incorrect answer 1", ..., "Incorrect answer N"],
"evidences": ["Text snippets from the article that supports the question and correct answer", "Another text snippet"]
}
# Knowledge Statement: {src_text}
Please provide your response in the specified format without any additional text.

*Figure 17.* Knowledge probing question synthesis template used in our experiments.

---

**EXAMPLE `src_text`**

"Hyperfine structure in EPR spectroscopy arises from interactions between unpaired electrons and nuclear spins."

---

**EXAMPLE OUTPUT**

{
"question": "What causes hyperfine structure in EPR spectroscopy?",
"correct_answer": "Interactions between unpaired electrons and nuclear spins",
"incorrect_answers": [
"Interactions between electron spins and lattice vibrations", "Coupling between electron orbitals and magnetic fields", "Dipolar interactions between neighboring nuclei", "Spin-orbit coupling within the electron cloud", "Chemical shift anisotropy of atomic orbitals" ],
"evidences": [
"Hyperfine structure in EPR spectroscopy arises from interactions between unpaired electrons and nuclear spins." ]
}

*Figure 18.* Knowledge probing question synthesis example input and output.

## F. LLM Usage Statement

We used GPT-o3 and GPT-5 from OpenAI for grammar and typo corrections.

*Table 15.* Domain-wise breakdown of SCIREAS tasks and instance counts (Part 1: Physics to Math).

| Domain | Task Source | Subtask/Subdomain | Instances | Total | Metrics |
|---|---|---|---|---|---|
| | GPQA | Physics | 187 | | Acc |
| | MMLU-Pro | physics | 200 | | Acc |
| | SciBench | fund | 81 | | Acc |
| | | thermo | 83 | | Acc |
| | | class | 63 | | Acc |
| | OlympiadBench-COMP | physics_en | 236 | | Acc |
| | SciKnowEval.L5 | physics_problem_solving | 200 | | LM |
| | SciEval | physics_knowledge_application | 29 | | Acc |
| | | physics_scientific_calculation | 200 | | Acc |
| | UGPhysics | Electrodynamics | 170 | | Acc |
| | | Thermodynamics | 200 | | Acc |
| Physics | | GeometricalOptics | 54 | 5087 | Acc |
| | | Relativity | 200 | | Acc |
| | | ClassicalElectromagnetism | 200 | | Acc |
| | | ClassicalMechanics | 200 | | Acc |
| | | WaveOptics | 200 | | Acc |
| | | QuantumMechanics | 200 | | Acc |
| | | TheoreticalMechanics | 200 | | Acc |
| | | AtomicPhysics | 200 | | Acc |
| | | SemiconductorPhysics | 148 | | Acc |
| | | Solid-StatePhysics | 154 | | Acc |
| | | StatisticalMechanics | 200 | | Acc |
| | SuperGPQA | Physics | 1482 | | Acc |
| | GPQA | Chemistry | 183 | | Acc |
| | MMLU-Pro | chemistry | 200 | | Acc |
| | SciBench | quan | 41 | | Acc |
| | | chemc | 47 | | Acc |
| | | atkins | 121 | | Acc |
| Chemistry | | matter | 57 | 2158 | Acc |
| | SciKnowEval.L5 | chemical_procedure_generation | 74 | | LM |
| | | chemical_reagent_generation | 125 | | LM |
| | SciEval | chemistry_knowledge_application | 200 | | Acc |
| | | chemistry_scientific_calculation | 200 | | Acc |
| | SuperGPQA | Chemistry | 910 | | Acc |
| | MMLU-Pro | computer science | 200 | | Acc |
| Comp Sci | SciRIFF | Qasper | 107 | 415 | F1, LM |
| | SuperGPQA | Computer Science and Technology | 108 | | Acc |
| | MMLU-Pro | math | 200 | | Acc |
| | SciBench | calc | 52 | | Acc |
| | | stat | 92 | | Acc |
| Math | | diff | 55 | 2533 | Acc |
| | OlympiadBench-COMP | maths_en | 674 | | Acc |
| | SuperGPQA | Mathematics | 1460 | | Acc |

*Table 16.* Domain-wise breakdown of SciReas tasks and instance counts (Part 2: Biology to Engineering).

| Domain | Task Source | Subtask | Instances | Total | Metrics |
|--------|-------------|---------|-----------|-------|---------|
| Biology | GPQA | Biology | 78 | | Acc |
| | MMLU-Pro | biology | 200 | | Acc |
| | LabBench | CloningScenarios | 33 | | Acc |
| | | ProtocolQA | 108 | | Acc |
| | | SeqQA | 600 | 1911 | Acc |
| | SciKnowEval.L5 | biological_procedure_generation | 200 | | LM |
| | | biological_reagent_generation | 200 | | LM |
| | SciEval | biology_knowledge_application | 200 | | Acc |
| | | biology_scientific_calculation | 200 | | Acc |
| | SuperGPQA | Biology | 92 | | Acc |
| Medicine | MMLU-Pro | health | 200 | | Acc |
| | SciRIFF | SciFact | 184 | 634 | F1, LM |
| | | Evidence Inference | 250 | | F1 |
| Material Sci | SciKnowEval.L5 | crystal_structure_and_composition | 196 | | LM |
| | | specified_band_gap_material_generation | 200 | 624 | LM |
| | | property_and_usage_analysis | 118 | | LM |
| | SuperGPQA | Materials Science and Engineering | 110 | | Acc |
| Engineering | MMLU-Pro | engineering | 200 | | Acc |
| | SuperGPQA | Control Science and Engineering | 77 | | Acc |
| | | Information and Communication Engineering | 156 | | Acc |
| | | Electrical Engineering | 234 | | Acc |
| | | Chemical Engineering and Technology | 226 | | Acc |
| | | Power Engineering and Engineering Thermophysics | 345 | 2205 | Acc |
| | | Electronic Science and Technology | 95 | | Acc |
| | | Hydraulic Engineering | 67 | | Acc |
| | | Mechanics | 456 | | Acc |
| | | Mechanical Engineering | 30 | | Acc |
| | | Civil Engineering | 93 | | Acc |
| | | Optical Engineering | 162 | | Acc |
| | | Nuclear Science and Technology | 30 | | Acc |
| | | Instrument Science and Technology | 12 | | Acc |
| | | Systems Science | 22 | | Acc |

*Table 17.* Domain-wise breakdown of SciReas-Pro tasks and instance counts (Part 1: Physics to Math).

| Domain | Task Source | Subtask/Subdomain | Instances | Total | Metrics |
|--------|-------------|-------------------|-----------|-------|---------|
| Physics | GPQA | Physics | 8 | | Acc |
| | MMLU-Pro | physics | 5 | | Acc |
| | SciBench | fund | 1 | | Acc |
| | | thermo | 10 | | Acc |
| | | class | 8 | | Acc |
| | OlympiadBench-COMP | physics_en | 25 | | Acc |
| | SciEval | physics_knowledge_application | 1 | | Acc |
| | | physics_scientific_calculation | 1 | | Acc |
| | UGPhysics | Electrodynamics | 17 | | Acc |
| | | Thermodynamics | 16 | | Acc |
| | | GeometricalOptics | 9 | 388 | Acc |
| | | Relativity | 16 | | Acc |
| | | ClassicalElectromagnetism | 21 | | Acc |
| | | ClassicalMechanics | 17 | | Acc |
| | | WaveOptics | 16 | | Acc |
| | | QuantumMechanics | 17 | | Acc |
| | | TheoreticalMechanics | 13 | | Acc |
| | | AtomicPhysics | 13 | | Acc |
| | | SemiconductorPhysics | 13 | | Acc |
| | | Solid-StatePhysics | 13 | | Acc |
| | | StatisticalMechanics | 15 | | Acc |
| | SuperGPQA | Physics | 133 | | Acc |
| Chemistry | GPQA | Chemistry | 31 | | Acc |
| | MMLU-Pro | chemistry | 3 | | Acc |
| | SciBench | quan | 3 | | Acc |
| | | chemc | 2 | | Acc |
| | | atkins | 6 | 135 | Acc |
| | | matter | 3 | | Acc |
| | SciEval | chemistry_knowledge_application | 11 | | Acc |
| | | chemistry_scientific_calculation | 3 | | Acc |
| | SuperGPQA | Chemistry | 73 | | Acc |
| Comp Sci | MMLU-Pro | computer science | 6 | 21 | Acc |
| | SuperGPQA | Computer Science and Technology | 15 | | Acc |
| Math | MMLU-Pro | math | 3 | | Acc |
| | SciBench | calc | 2 | | Acc |
| | | stat | 2 | 283 | Acc |
| | | diff | 3 | | Acc |
| | OlympiadBench-COMP | maths_en | 92 | | Acc |
| | SuperGPQA | Mathematics | 181 | | Acc |

*Table 18.* Domain-wise breakdown of SCIREAS-PRO tasks and instance counts (Part 2: Biology to Engineering).

| Domain | Task Source | Subtask | Instances | Total | Metrics |
|---|---|---|---|---|---|
| Biology | GPQA | Biology | 2 | | Acc |
| | MMLU-Pro | biology | 6 | | Acc |
| | LabBench | CloningScenarios | 2 | | Acc |
| | | ProtocolQA | 10 | 123 | Acc |
| | | SeqQA | 89 | | Acc |
| | SciEval | biology_knowledge_application | 3 | | Acc |
| | | biology_scientific_calculation | 2 | | Acc |
| | SuperGPQA | Biology | 9 | | Acc |
| Medicine | MMLU-Pro | health | 5 | 5 | Acc |
| Material Sci | SuperGPQA | Materials Science and Engineering | 13 | 13 | Acc |
| Engineering | MMLU-Pro | engineering | 14 | | Acc |
| | SuperGPQA | Control Science and Engineering | 7 | | Acc |
| | | Information and Communication Engineering | 15 | | Acc |
| | | Electrical Engineering | 32 | | Acc |
| | | Chemical Engineering and Technology | 43 | | Acc |
| | | Power Engineering and Engineering Thermophysics | 44 | 292 | Acc |
| | | Electronic Science and Technology | 13 | | Acc |
| | | Hydraulic Engineering | 13 | | Acc |
| | | Mechanics | 54 | | Acc |
| | | Mechanical Engineering | 7 | | Acc |
| | | Civil Engineering | 18 | | Acc |
| | | Optical Engineering | 23 | | Acc |
| | | Nuclear Science and Technology | 3 | | Acc |
| | | Instrument Science and Technology | 2 | | Acc |
| | | Systems Science | 4 | | Acc |

