# OpenReview forum: "Demystifying Scientific Problem-Solving in LLMs by Probing Knowledge and Reasoning"
_ICML.cc/2026/Conference — ICML 2026 regular_

### Official Review · Reviewer_iSnF · 2026-03-05

**Soundness:** 2
**Presentation:** 3
**Significance:** 3
**Originality:** 3
**Overall Recommendation:** 3
**Confidence:** 3

**Summary:**

This paper presents SCIREAS, a unified evaluation suite that brings together 10 existing benchmarks for scientific reasoning. It also introduces SCIREAS-PRO, a more challenging subset designed to separate complex reasoning from basic factual recall. In addition, the authors propose KRUX, an analytical probing framework that leverages a powerful language model to extract basic “knowledge ingredients” from reasoning chains and reinsert them into the model context. Using these tools, the authors conduct an empirical study showing that retrieving task-relevant knowledge from parametric memory remains a key bottleneck in scientific problem-solving. They further demonstrate that chain-of-thought fine-tuning improves models’ ability to activate this implicit knowledge.

**Compliance With Llm Reviewing Policy:**

Affirmed.

**Key Questions For Authors:**

- How exactly do you guarantee that the KI extraction prompt prevents the strong LLM from outputting intermediate reasoning steps that act as deductive hints for the target model?
- How would the performance of the base models augmented with KIs compare to base models augmented with top-k facts retrieved from a standard scientific corpus via RAG?

**Limitations:**

While this study verified the quality of KIs via manual review, it only evaluated randomly sampled reasoning traces. Such a sample size is relatively limited, and manual assessment inevitably carries subjective biases. Furthermore, the study did not establish standardized, quantifiable metrics for evaluating KI quality, which makes it hard to systematically and objectively regulate KI extraction in large-scale experiments.

**Strengths And Weaknesses:**

## Strengths:
- SCIREAS-PRO presents a particularly practical curation strategy. By filtering examples based on performance differences between models under high and low inference compute budgets, the authors are able to effectively target tasks that require genuine multi-step reasoning.
- The KRUX framework offers a convincing empirical approach for distinguishing between knowledge deficits and reasoning failures in model behavior.
- Their controlled post-training experimental setup is rigorously designed. This setup makes it possible to clearly attribute performance improvements to either better reasoning structures or the injection of domain knowledge.

## Weaknesses:
- The framework relies heavily on the premise that KIs strictly represent standalone facts. However, the operational definition of  KI is broad. In complex scientific problems, an intermediate deduction can masquerade as a "fact". If the KIs used in experiments encapsulate intermediate reasoning steps, then the KRUX pipeline is leaking reasoning structure, artificially inflating the performance.
- The entire probing pipeline is heavily dependent on DeepSeek-R1 as the KI extractor. The paper lacks a quantitative measure of consistency across different extraction models. If the extractor is biased toward certain phrasings or inadvertently summarizes the answer, the validity of the downstream probing is compromised.

---

> ### Author Rebuttal · Authors · 2026-03-31
>
> We appreciate the recognition on SciReas-Pro and KRUX.
> >W1: "… If the KIs used in experiments encapsulate intermediate reasoning steps, then the KRUX pipeline is leaking reasoning structure, artificially inflating the performance."\
> Q1: “How exactly do you guarantee that the KI extraction prompt prevents the strong LLM from outputting intermediate reasoning steps …?”
>
> Thank you for this important concern. Note that we provide the exact extraction prompt and examples in Appendix E.1. We address this concern with three lines of evidence:
>
> **(1) Rubric-based manual audit.** We expanded our audit from 100 to 300 reasoning traces across three different extractors (DeepSeek-R1, Qwen3-30B, Gemini-3-Flash), and each KI was checked against our rubric-based pipeline in Appendix E.2.1, line 1357-1358. Our result shows that all KIs from the three extractors “(a) do not refer to any specific identities in the question, […] (b) do not expose the final answers, and (c) adhere to the given traces with no additional information nor missing essential components.”
>
> **(2) Self-KI control.** Our self-KI baseline (where the model receives KIs from its own traces) shows no improvement, confirming that the extractor does not inject reasoning structure or additional knowledge beyond what the model already knows (lines 316-322). If KIs were leaking reasoning steps, self-KIs would also improve performance -- but they do not. This echoes with our manual check on KI extraction quality (Section 4.2 lines 309-314, Appendix E.2, and point (1)), confirming the extractor model does not impose additional knowledge beyond input reasoning traces during KI extraction. Also note that for all experiments with KIs, we randomly permute KIs in the prompt to mitigate potential structural hints.
>
> **(3) Domain-adjacent KI ablation.** We have added a set of experiments that mix each question's matched KIs with KIs from 5 other questions in the same subject from MMLU-Pro* (*SciReas version), and inject them with random permutations. If KIs were encoding intermediate reasoning steps specific to the original question, adjacent KIs should provide little benefit. Instead, adjacent KIs provide consistent improvements for both base and their reasoning variants (see our response to Reviewer n7Hz Q2), confirming that the benefit comes from generalizable domain knowledge rather than question-specific reasoning hints.
>
> The reviewer may argue that although KIs do not expose reasoning traces, providing a knowledge point alone could offer some hints at a potential search path to a solution. However, in KRUX setup, especially for RQ2, the exact same R1 KIs are provided to both the base models and their reasoning variants, meaning they share the same narrowed search space for a solution as a starting point. Therefore, we argue that our setup is fair for comparing the base models and their reasoning-fine-tuned variants, and our conclusion that knowledge can provide additional improvement on top of reasoning holds.
> >W2: “The entire probing pipeline is heavily dependent on DeepSeek-R1 as the KI extractor. The paper lacks a quantitative measure of consistency across different extraction models …”
>
> Our submitted manuscript already includes experiments using a second extractor (i.e., Qwen3-30B-A3B-Thinking-2507) and provides quantitative and qualitative analysis demonstrating that KRUX is __not__ specific to a certain extraction model (Appendix E.2). To further solidify this point, we have expanded the analysis with Gemini-3-Flash as the extractor and expand the manual audit for KI quality from 100 samples to 300 and confirmed KI extractors are reliable. Please see our response to Q1 from Reviewer n7Hz for more details.
> >Q2: “How would the performance of the base models augmented with KIs compare to base models augmented with top-k facts retrieved from a standard scientific corpus via RAG?”
>
> Thank you for your question. We did try using an E5 retriever with the Wikipedia18 corpus as an external knowledge base. However, the result does not show improvement (due to length constraints, we will add auxiliary results in the revised manuscript). The main reason in our observation is that the search system fails to retrieve highly relevant knowledge to solve reasoning-intensive questions. Even in the best query-doc matching case, a matched Wiki page might contain a lot of information that is not directly relevant to the question, whereas KIs extracted from LLMs provide atomic knowledge units for solving the problem. Proper indexing of existing retrieval corpora is a separate but active area of research, and our observation aligns with concurrent work [1][2]. Due to the scope of research, we chose to use a reliable extractor to derive such knowledge pieces.
>
> [1] Shao et al. 2025. ReasonIR: Training Retrievers for Reasoning Tasks. arXiv:2504.20595.
>
> [2] Lyu et al. 2025. Frustratingly Simple Retrieval Improves Challenging, Reasoning-Intensive Benchmarks. arXiv:2507.01297.

---

### Official Review · Reviewer_iRgr · 2026-03-12

**Soundness:** 4
**Presentation:** 4
**Significance:** 4
**Originality:** 3
**Overall Recommendation:** 5
**Confidence:** 4

**Summary:**

This paper addresses a central ambiguity in scientific problem solving with LLMs: when a model fails on a science task, is it because it lacks the necessary factual knowledge, or because it cannot effectively reason with that knowledge? The paper introduces SCIREAS, a unified suite of ten public science-focused benchmarks integrated into a standardized evaluation harness to enable reproducible analysis. It also presents SCIREAS-PRO, a reasoning-intensive subset, and KRUX, a probing framework that injects compact, answer-agnostic Knowledge Ingredients (KIs)—atomic facts extracted from another model’s CoT—to control for knowledge availability while measuring how target models actually use it.

The paper reports three main findings. First, base models augmented with high-quality in-context KIs often outperform their reasoning counterparts on science tasks. Second, reasoning models still benefit from additional external knowledge beyond the gains provided by improved reasoning alone. Third, KIs extracted from a math-reasoning-tuned model help a base model more than KIs extracted from the base model itself—even when both models already “know” the relevant facts—suggesting that improved recall and selection, rather than simply more knowledge, plays a key role.

**Compliance With Llm Reviewing Policy:**

Affirmed.

**Final Justification:**

My concerns have been addressed.

**Key Questions For Authors:**

See Weaknesses.

**Limitations:**

yes

**Strengths And Weaknesses:**

# Strengths
**Soundness**

1. The KRUX framework provides a reasonable experimental design to control for knowledge availability by injecting answer-agnostic Knowledge Ingredients (KIs), allowing clearer measurement of how models utilize knowledge.

2. Empirical results across SCIREAS benchmarks consistently support the main claims, particularly the large performance gains (≥10%) when supplying high-quality KIs.

**Presentation**

1. The paper is clearly written and easy to follow. The motivation, benchmark construction (SCIREAS / SCIREAS-PRO), and KRUX framework are presented in a logical and understandable way.

**Significance**

1. The paper addresses an important question in LLM evaluation: distinguishing whether failures come from missing knowledge or weak reasoning.

2. Results highlight that retrieval/memory and reasoning are complementary, which provides useful insights for designing future LLM systems.

**Originality**

1. The KRUX framework offers a creative way to probe reasoning by controlling knowledge through injected Knowledge Ingredients.

2. The SCIREAS benchmark suite provides useful infrastructure for evaluating science reasoning tasks.

# Weaknesses

**Soundness**
1. The selection of SCIREAS-PRO relies on proprietary models, which may introduce model-specific biases. It is unclear whether the filtering results would be consistent with open-source models.


**Presentation**
1. The paper could further clarify how KIs are extracted and how consistent or diverse the extracted KIs are across different source models.


**Significance**
1. Experiments mainly focus on science benchmarks, leaving open the question of how well the conclusions generalize to other domains.


**Originality**
1. Some components build on existing ideas such as extracting information from CoT traces and injecting external context, so the novelty mainly lies in the evaluation design rather than new modeling techniques.

---

> ### Author Rebuttal · Authors · 2026-03-31
>
> We are grateful for the strong assessment and the recognition.
>
> > W-Soundness: "The selection of SCIREAS-PRO relies on proprietary models, which may introduce model-specific biases. It is unclear whether the filtering results would be consistent with open-source models."
>
> Thank you for your question. In response to the concern of overreliance on proprietary models for the selection of SciReas-Pro, we construct SciReas-Pro-Qwen, where the instances are selected by picking instances that Qwen3-32B answered correctly with thinking mode ON and answered incorrectly with thinking mode OFF. The process ends up with 3,208 examples, and the results show that the performance gap while evaluating on SciReas-Pro-Qwen with different reasoning efforts is significantly larger than the gap indicated by SciReas across open and closed models. This effectively shows that the filtering method is **not model-idiosyncratic** and can be applied with open source models, preserving the feature as an indicator of the amplified performance gap between different reasoning intensities.
>
> | Level | o3: SciReas | o3: SciReas-Pro-Qwen | Claude-4-Sonnet: SciReas | Claude-4-Sonnet: SciReas-Pro-Qwen | DeepSeek-v3/R1: SciReas | DeepSeek-v3/R1: SciReas-Pro-Qwen | GPT-OSS: SciReas | GPT-OSS: SciReas-Pro-Qwen |
> |---|---:|---:|---:|---:|---:|---:|---:|---:|
> | Low | 66.2 | 73.4 | 60.3 | 55.1 | 51.7 | 40.3 | 59.1 | 54.3 |
> | High | 68.4 | 77.8 | 62.1 | 62.3 | 64.2 | 73.6 | 61.3 | 61.0 |
> | Gap | +2.2 | **+4.4** | +1.8 | **+7.2** | +12.5 | **+33.3** | +2.2 | **+6.7** |
>
> > W-Presentation: "The paper could further clarify how KIs are extracted and how consistent or diverse the extracted KIs are across different source models."
>
> To clarify, we present detailed procedures, including the specific prompt we used in our manuscript (Appendix E.1. Figure 12). For alternative extractors, we provide qualitative and quantitative analysis on using Qwen3-30B-A3B-Thinking-2507 vs DeepSeek-R1 as the extractor (Appendix E.2). To further address reviewers’ concern, we add experiments using Gemini-3-Flash as another extractor (please see response to Q1 from Reviewer n7Hz). We will update our presentation in the manuscript.
>
> > W-Significance: "Experiments mainly focus on science benchmarks, leaving open the question of how well the conclusions generalize to other domains."
>
> We thank the reviewer for the comment. We’d like to clarify that our focus on scientific problem-solving is deliberate. Science is a particularly suitable setting for our study because these tasks require both substantial domain knowledge and complex reasoning, which is exactly the interaction we aim to analyze. This question is important because scientific problem-solving is a central target application for frontier LLMs, and progress in this setting depends on understanding whether failures come from missing knowledge, poor knowledge retrieval, or reasoning limitations. Expert-level scientific benchmarks therefore provide an appropriate stress test for studying the factors that drive model performance and how they interact.
>
> > W-Originality: "Some components build on existing ideas such as extracting information from CoT traces and injecting external context, so the novelty mainly lies in the evaluation design rather than new modeling techniques."
>
> We appreciate this observation and agree that our primary contribution is methodological and analytical rather than architectural. KRUX provides a principled, reusable framework for studying and understanding the distinct roles of reasoning vs. domain knowledge in LLMs. The findings it enables (that knowledge retrieval is a critical bottleneck in Section 4.3, that reasoning and knowledge benefits are additive in Section 4.4, and that reasoning fine-tuning improves knowledge surfacing in Section 4.5) provide actionable insights for system design that go beyond any single modeling technique.

---

> > ### Author Rebuttal · Reviewer_iRgr · 2026-04-02
> >
> > Thank you for your response. My concerns have been addressed.

---

> > > ### Author Response · Authors · 2026-04-07
> > >
> > > Thank you for your comments! We will incorporate the update in our revised manuscript.

---

### Official Review · Reviewer_7ZPk · 2026-03-12

**Soundness:** 3
**Presentation:** 2
**Significance:** 2
**Originality:** 3
**Overall Recommendation:** 5
**Confidence:** 4

**Summary:**

This paper introduces SCIREAS, a benchmark comprising diverse scientific reasoning tasks designed to investigate the impact of models' internal knowledge and their reasoning capabilities in scientific problem-solving. They propose the KRUX framework, which includes a knowledge-augmented data pipeline and training for models. Through comprehensive analysis across multiple models, the paper presents valuable insights into the interplay between knowledge and reasoning.

**Compliance With Llm Reviewing Policy:**

Affirmed.

**Final Justification:**

The author addressed my concerns, and I will raise my score.

**Key Questions For Authors:**

See comments in Strengths and Weaknesses.

**Limitations:**

1. The details of data construction need to be clarified.

2. Some important experiments and case studies are missing.

3. The performance gains from simple SFT appear marginal. The authors could consider reinforcement learning approaches combined with KRUX's data augmentation strategy, which might yield more significant findings.

**Strengths And Weaknesses:**

**Strengths**

This paper presents SCIREAS, a benchmark encompassing diverse scientific reasoning tasks to disentangle the effects of internal knowledge and reasoning capabilities in LLMs. The proposed KRUX framework features a knowledge-augmented construction pipeline.  Through experiments across multiple models and different training settings, the paper yields insightful findings regarding knowledge-reasoning interactions.

**Weaknesses**

1. More details on SCIREAS data filtering and construction should be provided. While I understand page limitations, critical information currently relegated to the appendix should at least be summarized in the main text, such as the names of the 10 original source benchmarks, dataset statistics, and specific LLMs used.

2. The data filtering process mentions sampling from high-cost benchmarks like MMLU-Pro. The authors should clarify whether this sampling step occurs before or after the manual screening and knowledge scope annotation stage.

3. The paper lacks experiments validating whether KRUX actually improves performance on SCIREAS. Currently, only standard SFT results are presented. Moreover, the connection between KRUX and SCIREAS appears weak. They read like two separate contributions rather than an integrated framework.

4. Experimental results show that Qwen w/ Qwen KIs performs significantly worse than direct SFT, sometimes even below the base model. The authors should explain this phenomenon, which also appears in the Llama experiments.

5. Case studies are insufficient. For example, during SCIREAS construction, it would be helpful to see qualitative examples illustrating how questions were filtered, transformed, or annotated with knowledge scopes. In addition, different training settings for KRUX can be used as case studies to analyze what mainly affects the performance gain.

If the authors address my concerns, I would consider revising my score.

---

> ### Author Rebuttal · Authors · 2026-03-31
>
> We appreciate the recognition of SciReas and KRUX.
> > W1: “More details on SCIREAS data filtering and construction should be provided. While I understand page limitations, critical information currently relegated to the appendix should at least be summarized in the main text …” \
> L1: “The details of data construction need to be clarified.”
>
> We thank the reviewer for their constructive feedback and understanding. We list our 3-step exclusion protocol and validation procedure for dataset construction in Appendix B.1 with examples associated with each step, along with the reasoning for the tasks selected from each benchmark. We also listed the names of 10 selected datasets in the main body Section 2 lines 113-121. We will consolidate and summarize important details in Section 3.1 as we update our manuscript.
> > W2: “... The authors should clarify whether this sampling step occurs before or after the manual screening and knowledge scope annotation stage.”
>
> Thank you for the question. To clarify, sampling occurs *after* manual screening and knowledge scope annotation, so it does not interfere with screening procedures. The analysis on sampling validation in Appendix B.2 is also conducted after manual screening. We will emphasize this when we introduce SciReas in Section 3.1 on the revised manuscript.
> > W3: “… Currently, only standard SFT results are presented …”\
> W5: “… In addition, different training settings for KRUX can be used as case studies to analyze what mainly affects the performance gain.”\
> L3: “… The authors could consider reinforcement learning approaches combined with KRUX's data augmentation strategy, which might yield more significant findings.”
>
> We thank the reviewer for their comment. The goal of Section 4 is *not* to provide an exhaustive comparison of post-training algorithms, but to run *controlled interventions* that isolate how reasoning and domain knowledge each affect scientific reasoning behavior. SFT is chosen because it enables clean and controlled comparisons, in pair with data level intervention (Math, STEM, BOTH) to study how different knowledge and reasoning signals interact. We also note that we do not ignore RL-based post-training in evaluation: Table 1 already includes concurrent methods trained with RL or SFT+RL, including General-Reasoner and Llama-Nemotron. Our SFT checkpoints are competitive with these baselines and are therefore suitable as controlled checkpoints for the KRUX analysis.
> > W3: “The paper lacks experiments validating whether KRUX actually improves performance on SCIREAS … Moreover, the connection between KRUX and SCIREAS appears weak …”
>
> SciReas provides the unified and reasoning-intensive evaluation, while KRUX is the *probing framework* applied on top of this evaluation to diagnose reasoning, knowledge recall, and knowledge use. We will make this dependency clearer in the Introduction and at the beginning of Section 4.
> > W4: “Experimental results show that Qwen w/ Qwen KIs performs significantly worse than direct SFT, sometimes even below the base model …”
>
> Thank you for raising this point. When a model is fed KIs extracted from its own reasoning traces (i.e., self KIs), we expect *no* improvement. This self KI baseline verifies that the KI extractor faithfully isolates information already present in the model's reasoning, without hallucinating new knowledge or leaking answers. This echoes with our manual check on KI extraction quality (Section 4.2 lines 309-314 and Appendix E.2), confirming the extractor model **does not impose additional knowledge beyond input reasoning traces** during KI extraction.
>
> If self-KIs *did* improve performance, it would undermine our experimental design by suggesting that either (a) additional context tokens alone drive gains, or (b) the extractor is injecting its own knowledge during extraction. The fact that self-KIs show no improvement (or slight degradation due to added prompt length), while R1 KIs show large gains (+10% or more), is precisely what validates our framework as it confirms that the performance gains from R1 KIs are attributable to genuinely new knowledge, not extraction artifacts.
> > W5: “Case studies are insufficient. For example, during SCIREAS construction, it would be helpful to see qualitative examples illustrating how questions were filtered, transformed, or annotated with knowledge scopes …”\
> L2: “Some important experiments and case studies are missing.”
>
> We thank the reviewer for their comments and suggestions.
> For SciReas construction, we provide examples for each step in our subtask selection protocol in Appendix B.1 (lines 832-836, lines 842-846 and lines 848) as well as a cross-validation analysis in lines 851-858.
>
> For KRUX, we provide case studies regarding the alternative extractor (Appendix E.2, Table 11-12), qualitative and quantitative analysis on KI quality, as well as additional analysis in response to W1 and Q2/Q3 from Reviewer n7Hz.
> We would be happy to answer any additional questions.

---

> > ### Author Rebuttal · Reviewer_7ZPk · 2026-04-03
> >
> > Thank you for your reply! I will raise my score.

---

> > > ### Author Response · Authors · 2026-04-07
> > >
> > > Thank you for your comments and recognition! We will incorporate the update in our revised manuscript.

---

### Official Review · Reviewer_n7Hz · 2026-03-13

**Soundness:** 3
**Presentation:** 4
**Significance:** 3
**Originality:** 3
**Overall Recommendation:** 5
**Confidence:** 4

**Summary:**

The paper introduces SciReas which is a evaluation framework consolidating 10 existing scientific benchmarks under a standardized harness, and SciReas-Pro, a reasoning-intensive subset. Further, they proposes KRUX, a probing framework that studies a controlled separation of knowledge retrieval from reasoning capability. The authors identify three findings: (1) knowledge retrieval is a key bottleneck in scientific reasoning, (2) reasoning-fine-tuned models can still improve with external knowledge, and (3) reasoning fine-tuning improves knowledge surfacing. The paper supports these findings with experiments on Qwen2.5-7B and Llama-3.1-8B.

**Compliance With Llm Reviewing Policy:**

Affirmed.

**Final Justification:**

The authors provide experimental responses to the main concerns. The adjacent-KI experiment directly responds to the hint-leakage concern that the gains persist even when KIs are mixed with same-subject but question-irrelevant knowledge, supporting that improvements stem from genuine knowledge availability rather than solution-path hints. The authors further acknowledge the limitations.

**Key Questions For Authors:**

1. How can we be certain that the observed performance gains stem from genuine knowledge utilization rather than the model simply following KI driven solution paths?
2. Regarding RQ2, the performance gains might reflect the model's superior ability to exploit the structural hints leaked by the KIs. Have the authors considered an ablation using KIs from domain-adjacent questions to differentiate true reasoning from hint-following?
3. Have the authors compared KI injection against simply providing relevant domain knowledge from an external source at inference time?

**Limitations:**

While the authors briefly acknowledge potential societal impacts, the paper lacks an explicit discussion of methodological limitations. SciReas-Pro and KI extraction both depend on proprietary models which can introduce inherent biases and reproducibility issues.

**Strengths And Weaknesses:**

**Strengths**
1. The experiment design is simple and well motivated. It successfully operationalizes the distinction between knowing a fact but failing to retrieve it versus simply not knowing it at all.
2. RQ1 and RQ2 present important findings.  RQ1 validates that a base model given KIs outperforms reasoning-fine-tuned models without KIs by over 10% which is then explained in RQ3 that the knowledge was already in the base model and reasoning fine-tuning just makes it better at surfacing it.
3. SciReas standardizes 10 heterogeneous benchmarks under one harness with consistent prompting and scoring which is a timely and valuable contribution to the community.

**Weaknesses**
1. KI extraction is validated on only 100 manually checked samples but raises concern as they are not raw knowledge but instead R1's capabilities to judge which facts are relevant which makes KI highly dependent on R1's retrieval capability.
2. Since KIs are extracted from the exact same question being evaluated, the specific set of selected KIs may hin the solution path. This makes it hard to understand if gains come from knowledge availability or instead from the hints about how to approach the problem.
3. RQ2 attempts to establish that knowledge and reasoning are complementary. But if KIs leak hints on the potential path to solution, reasoning-finetuned models may simply be better at exploiting those hints rather than genuinely reasoning on top of the knowledge.

---

> ### Author Rebuttal · Authors · 2026-03-31
>
> We appreciate the positive assessment of our experiment design and for noting important findings.
> >W1: "KI extraction is validated on only 100 manually checked samples but raises concern as they are not raw knowledge but instead R1's capabilities to judge which facts are relevant which makes KI highly dependent on R1's retrieval capability.”
>
> We address the concern of extractor dependency and KI quality by (1) presenting experiment results of alternative extractors, and (2) expanding qualitative manual inspection of KI quality.
>
> For (1), Appendix E.2 already reports results with Qwen3-30B-A3B-Thinking-2507 as extractor; we additionally include Gemini-3-Flash. We highlight GPQA results below. Across all extractors, R1 KIs in context (w/ R1 KIs) substantially outperform self KIs for both base and reasoning-enhanced models, demonstrating KRUX is **not** constrained to a specific extractor.
> ||**Qwen3**|||**DeepSeek-R1**|||**Gemini-3-Flash**|||
> |---|---|---|---|---|---|---|---|---|---|
> |**Setup**|w/self KIs|w/R1 KIs|Δ|w/self KIs|w/R1 KIs|Δ|w/self KIs|w/R1 KIs|Δ|
> |Qwen|35.0|43.5|+8.5|34.2|47.2|+13.0|35.0|46.7|+11.7|
> |Qwen-STEM|42.6|51.1|+8.5|41.6|52.5|+10.9|42.2|54.0|+11.8|
> |Qwen-MATH|38.8|50.2|+11.4|39.5|53.5|+14.1|40.6|51.6|+11.0|
> |Qwen-BOTH|43.1|52.9|+9.8|40.8|54.5|+13.7|44.9|52.2|+7.3|
> |||
> |Llama|29.0|39.5|+10.5|29.1|43.6|+14.5|29.5|42.0|+12.5|
> |Llama-STEM|38.6|51.8|+13.2|39.0|53.2|+14.2|40.8|55.4|+14.6|
> |Llama-MATH|33.9|50.0|+16.1|36.2|53.8|+17.6|39.1|54.9|+15.8|
> |Llama-BOTH|39.1|52.2|+13.2|39.4|54.7|+15.3|43.3|56.7|+13.4|
>
> For (2), according to our rubric-based analysis for KI quality in Appendix E.2.1 lines 1357-1358, we manually inspect additional 200 examples (300 in total, from three different extractors). The results remain that *all* KIs “(a) do not refer to any specific identities in the question, […] (b) do not expose the final answers, and (c) adhere to the given traces with no additional information nor missing essential components.”
> > W2: “… the specific set of selected KIs may hint the solution path. This makes it hard to understand if gains come from knowledge availability or instead from the hints about how to approach the problem.”\
> W3: “… if KIs leak hints on the potential path to solution, reasoning-finetuned models may simply be better at exploiting those hints rather than genuinely reasoning on top of the knowledge.”\
> Q1: “How can we be certain that the observed performance gains stem from genuine knowledge utilization rather than the model simply following KI driven solution paths?”\
> Q2: “Regarding RQ2, the performance gains might reflect the model's superior ability to exploit the structural hints leaked by the KIs …”
>
> The reviewer raises an important distinction (discussed in Section 4.4, lines 374-378). While KIs do not expose final answers, we expect they also identify the most relevant knowledge, making reasoning easier (e.g., a theorem proof becomes easier if told which lemmas to use, even if we already knew those lemmas beforehand). In our experiments, we apply multiple permutations on KIs for each run to mitigate this effect (lines 370-372), but we expect that the in-context KIs provide this kind of knowledge-relevance signal and that this is part of their power.
>
> As noted in the paper (lines 369-378), for RQs 1 and 2, we do not disentangle these two factors (supplying knowledge vs. identifying relevant knowledge). Instead, we show that relevant in-context knowledge aids both base and reasoning models. Since the same R1 KIs are provided to both variants, the comparison is fair. We also provide additional experiments using adjacent KIs as proposed in Q2 (below) to further mitigate knowledge-relevance signals. Further disentangling the two factors would require substantial additional experiments, and we leave it as an item of future work. We will expand our discussion of this issue in our revisions.
> > Q2: “... Have the authors considered an ablation using KIs from domain-adjacent questions to differentiate true reasoning from hint-following?”
>
> We appreciate the thoughtful suggestions. We created adjacent KI sets for each MMLU-Pro* (*SciReas version) question by mixing its KIs with KIs from 5 other same-subject questions (categorized by MMLU-Pro), injected with random permutations. Both base and reasoning models consistently improve, confirming gains come from added knowledge and that KRUX derived KIs are answer-agnostic.
> |**Setup**|w/self KIs|w/adjacent KIs|Δ|
> |---|---|---|---|
> |Qwen|59.0|64.9|+5.9|
> |Qwen-STEM|64.7|69.3|+4.6|
> |Qwen-MATH|66.9|71.6|+4.7|
> |Qwen-BOTH|65.7|71.0|+5.3|
> |||
> |Llama|47.7|58.1|+10.4|
> |Llama-STEM|59.1|64.3|+5.2|
> |Llama-MATH|59.7|63.3|+3.6|
> |Llama-BOTH|63.8|69.1|+5.3|
> > Q3: “Have the authors compared KI injection against simply providing relevant domain knowledge from an external source at inference time?”
>
> We tried retrieving relevant content from Wikipedia pages, but it proved unhelpful. Please see our response to Q2 from Reviewer iSnF for more details.

---

> > ### Author Rebuttal · Reviewer_n7Hz · 2026-04-04
> >
> > I would raise the score as the answers address the raised concerns.

---

> > > ### Author Response · Authors · 2026-04-07
> > >
> > > Thank you for your comments and recognition! We will incorporate the update in our revised manuscript.

---

### Decision · Program_Chairs · 2026-04-30

**Decision:**

Accept (regular)

**Comment:**

The paper presents **SciReas**, a standardized harness for 10 scientific benchmarks, along with **SciReas-Pro**, a subset focused on reasoning-intensive tasks. Additionally, it introduces **KRUX**, a probing framework that injects "Knowledge Ingredients" (KIs) into the model context to disentangle the bottlenecks of factual knowledge recall versus logical reasoning.

The reviewers initially expressed several concerns, which were addressed through extensive new experiments and clarifications:
* **Knowledge Leakage:** Reviewers **n7Hz** and **iSnF** were concerned that KIs might leak the "solution path" or reasoning structure rather than just factual knowledge. The authors responded with a **Domain-Adjacent KI** experiment, mixing relevant KIs with irrelevant facts from the same domain. The performance gains persisted, suggesting the models were using the knowledge rather than simply following hints.
* **Extractor Dependency:** There were concerns regarding the reliance on DeepSeek-R1 for KI extraction. The authors provided additional results using **Qwen3**, **Gemini-3-Flash**, and **GPT-4-o3**, showing that the findings are robust across different extractor models.
* **Probing Framework Validation:** The authors clarified the role of "self-KIs" (KIs extracted from the model's own traces). The fact that self-KIs provide no improvement (verifying no added knowledge or structural leakage) while R1-KIs provide a ~10% gain validates KRUX as a factual recall probe.

The paper addresses a fundamental question in scientific problem-solving: whether failures stem from "not knowing" or "not reasoning." The experimental design is rigorous, and the findings provide actionable insights for LLM development. The standardized benchmark (SciReas) is a timely contribution to the scientific AI community.